# GLOBAL CONTEXT-AWARE REPRESENTATION LEARNING FOR SPATIALLY RESOLVED TRANSCRIPTOMICS

## ABSTRACT

Spatially Resolved Transcriptomics (SRT) is a cutting-edge technique that captures the spatial context of cells within tissues, enabling the study of complex biological networks. Recently, graph-based deep learning has been utilized in identifying meaningful spatial domains by leveraging both gene expression and spatial information. However, these approaches fall short in obtaining qualified spot representations, particularly for those located around the boundary of spatial domains, as they heavily emphasize spatially local spots that have minimal feature differences from an anchor node. To address this limitation, we propose a novel framework, Spotscape, which introduces the Similarity Telescope module designed to learn spot representations by capturing the global relationships among multiple spots. Additionally, to address the challenges that arise when integrating multiple slices from heterogeneous sources, we propose a similarity scaling strategy that explicitly regulates the distances between intra- and inter-slice spots to ensure they remain nearly the same. Extensive experiments demonstrate the superiority of Spotscape in various downstream tasks, including spatial domain identification, multi-slice integration, and alignment tasks, compared to baseline methods. Our code is available at the following link: https://anonymous.4open.science/r/Spotscape-E312/

## 1 INTRODUCTION

Recently, Spatially Resolved Transcriptomics (SRT) has gained significant attention for its ability to capture the spatial context of cells within tissues. Specifically, advanced SRT technologies such as 10x Visium (Maynard et al., 2021), 10x Xenium (Janesick et al., 2023), seqFISH (Lubeck et al., 2014), and Stereo-seq (Chen et al., 2022a) provide spatially resolved gene expression data. These datasets not only contain gene expression profiles, which quantify the activity levels of thousands of genes within each spot of tissue, but also include spatial coordinates, which represent the exact physical location of each spot within the tissue. Since much of SRT data analysis focuses on specific spatial regions or their interactions, spatial domain identification (SDI) serves as a crucial initial step for categorizing distinct, biologically meaningful tissue regions. For this reason, initial studies typically employ unsupervised clustering methods (Blondel et al., 2008; Wolf et al., 2018b; Hao et al., 2021) to group spots based on their original gene expression data. However, they fall short in predicting accurate domain identification results due to the inherent noise in SRT data, which arises from the limited resolution of the technology, and the high dimensionality of the data.

In response to these challenges, various deep representation learning methods have been proposed to learn spot representations that capture biologically meaningful content by leveraging both spatial and gene expression data. Specifically, graph-based methods such as SEDR (Xu et al., 2024) and SpaGCN (Hu et al., 2021) construct graphs based on spatial coordinates to gather information from nearby spots and generate representations using graph neural networks (GNNs). While this approach effectively incorporates spatial information into latent representations, it has limitations, particularly for spots located around the boundary of different spatial domains. These boundary spots may receive information from nodes representing different types of spots (i.e., heterophilic nodes), which can complicate accurate representation learning.

To address this limitation, STAGATE (Dong & Zhang, 2022) proposed leveraging graph attention networks (GAT) (Veličković et al., 2017) to learn similarities between spots without solely depending on pre-defined edge weights, thereby enhancing the representations of spots at the boundaries of

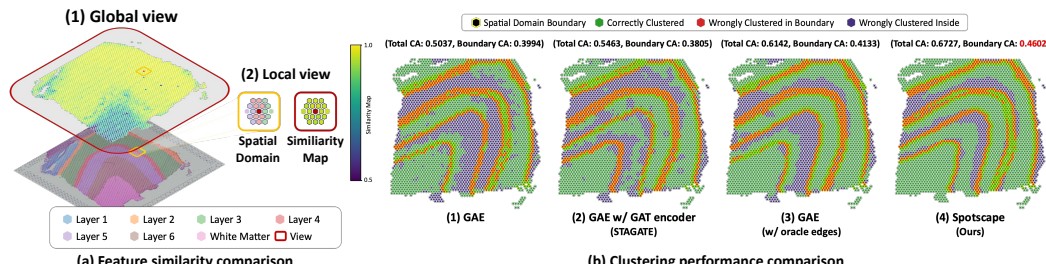

Figure 1: (a) Feature similarity comparison from global and local perspectives. In global view, the similarity between the anchor (i.e., red dot) and other spots gradually changes with their spatial coordinates. In contrast, in the local view, neighboring spots exhibit minimal feature discrepancy compared to the anchor, irrespective of the true spatial domain. (b) Clustering performance comparison in terms of clustering accuracy for all spots (Total CA) and particularly for spots located at the boundary of clusters (Boundary CA) in the human dorsolateral prefrontal cortex (DLPFC) dataset.

spatial domains. Despite the effectiveness of STAGATE, we argue that learning attention weights in the SRT data is particularly challenging due to the *continuous nature* of biological systems, where gene expression values tend to vary smoothly along spatial coordinates (Cembrowski & Menon, 2018; Phillips et al., 2019; Adler et al., 2019; Harris et al., 2021). This inherent continuity can, in some cases, complicate the distinction between different spatial domains (See Figure 1 (a)). Moreover, even if a model successfully assigns appropriate edge weights (e.g., high weights between spots of the same type and low weights otherwise), an anchor spot cannot obtain useful information from its neighboring spots due to the small feature discrepancies between the anchor and its neighboring spots. To corroborate our argument, in Figure 1 (b), we compared clustering performance of various graph autoencoder (GAE) architectures: (1) GAE on the original spatial nearest neighbor (SNN) graph[1], (2) GAE with a GAT encoder, (3) GAE with oracle edge weights[2], and (4) GAE that incorporates global similarity learning (our proposed method). We observe that while the attention mechanism is helpful for improving the general clustering performance (i.e., Total CA), it rather degrades the clustering performance of boundary spots (i.e., Boundary CA). This highlights the difficulty of learning spot representations near the boundary of spatial domains using attention. Another interesting observation is that even with oracle edge weights, improvements in terms of boundary CA is not significant compared with the GAE on the original SNN, supporting our argument that solely relying on the local view provides limited information.

In addition to addressing the aforementioned challenges in the single-slice analysis, representation learning models for the SRT dataset must account for batch effects Li et al. (2020b) to enable multi-slice analysis in the SRT data. Note that the batch effect refers to the phenomenon where spot representations from the same slice are unexpectedly clustered together regardless of their biological relevance, when integrating multiple datasets from different slices. While integrating multiple datasets offers significant advantages, addressing batch effects remains a key challenge.

To this end, we propose a novel framework, Spotscape, designed to address challenges in both the single-slice and multi-slice tasks, including Spatial Domain Identification (SDI) (i.e., single-slice task), SRT data integration and alignment (i.e., multi-slice task). To address our findings that exploring only spatially local neighbors yields limited performance gains, Spotscape introduces the Similarity Telescope module, which reflects the relative similarity not only among spatially neighboring nodes but also across global spots. More precisely, Spotscape generates two augmented views from the SNN graph and minimizes the difference between similarities calculated based on the two augmented views to preserve the meaningful similarities in the global context. This learning scheme is particularly beneficial for SRT data, as optimizing similarity is closely related to the clustering task, which is the most important downstream application. Moreover, Spotscape utilizes the prototypical contrastive loss, which groups semantically similar representations together while distancing dissimilar ones, resulting in fine-grained representations. This characteristic is particularly beneficial for addressing challenges that require more detailed representations, such as capturing

---

[1]The SNN graph is constructed by connecting spots that are either within a predefined radius $r$ or among the nearest top $k$ neighbors based on spatial distance.

[2]Edges between spots of the same type were assigned a weight of 1, and 0 otherwise. That is, we remove heterophilic edges.

rare cell types. Furthermore, we extend Spotscape to multi-slice tasks by addressing batch effects through a similarity scale matching strategy that explicitly balances the similarity scales of inter- and intra-relationships. This approach enables the effective mixing of representations across different slices, enabling our model applied to both single and multi-slice SRT data.

In summary, our contributions are four-fold:

- We discover that learning similarity between spatially local neighbors is insufficient for learning representations in the SRT data, especially near the boundary of spatial domains.
- To address this limitation, we propose a global similarity learning scheme called the Similarity Telescope module to capture the relationships between spots in the global context and adopt prototypical contrastive learning scheme, which helps the model to learn fine-grained representations in the SRT data.
- We propose a similarity scale matching strategy to address batch effects that arise when training multiple slices simultaneously, enabling our model to be effectively applied to both single-slice and multi-slice SRT data.
- We conduct extensive experiments in spatial domain identification, slice integration, and slice alignment to validate the superiority of Spotscape.

## 2 RELATED WORK

### 2.1 SPATIAL DOMAIN IDENTIFICATION

Spatial domain identification (SDI) is crucial for categorizing biologically meaningful tissue regions and advancing understanding of transcriptional structures, spatial heterogeneity, and cell interactions, thereby aiding insights into tissue organization (Maynard et al., 2021), disease progression (Chen et al., 2022b), and targeted therapies (Maynard et al., 2021; Chen et al., 2022b; Arora et al., 2023). To improve upon traditional clustering methods (Blondel et al., 2008; Wolf et al., 2018b; Hao et al., 2021) used in single-cell RNA sequencing, Giotto (Dries et al., 2021) and BayesSpace (Zhao et al., 2021) leverage hidden Markov random fields and Bayesian techniques, respectively, to incorporate spatial data. Recently, graph-based deep learning methods have emerged to jointly use spatial coordinates and gene expression. For instance, SEDR (Xu et al., 2024) employs a graph autoencoder with masking to learn and denoise spatial gene expression, while SpaGCN (Hu et al., 2021) uses graph neural networks (GNNs) and clustering loss (Xie et al., 2016) for integration of spatial information and gene expression. STAGATE (Dong & Zhang, 2022) applies graph attention networks (GAT)(Veličković et al., 2017) to address boundary heterogeneity. Moreover, self-supervised learning has become popular for capturing robust representations without labels; SpaceFlow(Ren et al., 2022) uses Deep Graph Infomax (DGI)(Veličković et al., 2018) with spatial regularization for spatial consistency, and SpaCAE(Hu et al., 2024) utilizes a graph autoencoder with contrastive learning to handle sparse and noisy spatially resolved transcriptomics (SRT) data effectively.

### 2.2 SLICE INTEGRATION AND ALIGNMENT

Numerous SRT studies collect data from neighboring tissue sections, but inconsistencies in how the slices are dissected and positioned on the array result in misaligned spatial coordinates. As a result, combining data across different slices is a complex yet essential task to extract diverse and valuable insights. To address this, PASTE Zeira et al. (2022) uses an optimal transport approach to align the spots and integrate them into a shared embedding space. Additionally, SRT data is sometimes generated under varying conditions, such as different technology platforms, developmental stages, or sample conditions. We refer to this as the heterogeneous case, which presents an additional challenge: batch effects, where spot representations from the same slice cluster together, irrespective of their biological significance. To overcome this, STAligner Zhou et al. (2023) defines mutual nearest neighbors as positive samples and utilizes the triplet loss to reduce the distance between anchor and positive samples, facilitating the integration of embeddings across different slices. In addition, GraphST (Long et al., 2023) leverages DGI (Veličković et al., 2018) to maximize mutual information of spots from vertical or horizontal integration to correct batch effect. Moreover, SLAT (Xia et al., 2023) employs a graph adversarial training scheme for robustly aligning spatial slices. Our approach addresses both homogeneous and heterogeneous integration and alignment tasks using a simple similarity scale matching strategy.

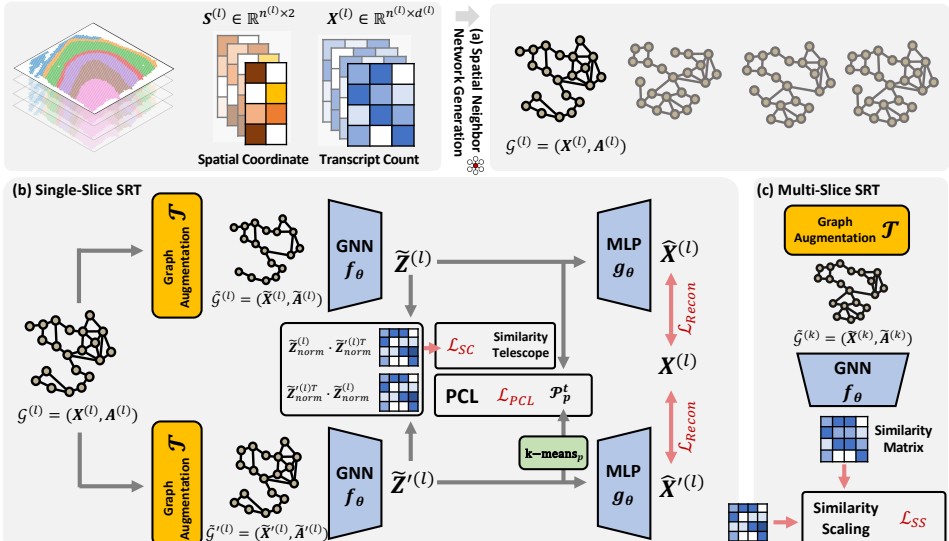

Figure 2: Overall framework of Spotscape. (a) Given SRT data composed of spatial coordinates and transcript counts, we construct a spatial nearest neighbor (SNN) graph. The model is then trained with the SNN graph using (b) similarity telescope and PCL loss, while additionally utilizing (c) similarity scaling loss in multi-slice SRT.

## 3 PROBLEM STATEMENT

**Notations.** Given the SRT data composed of spatial coordinates $S \in \mathbb{R}^{N_s \times 2}$ and gene expression profile $X \in \mathbb{R}^{N_s \times N_g}$, where $N_s$ represents number of spots and $N_g$ the number of genes, we construct a spatial nearest neighbors (SNN) graph $\mathcal{G} = (X, A)$ based on distance calculated by spatial coordinates. The adjacency matrix $A \in \mathbb{R}^{N_s \times N_s}$ is defined such that $A_{ij} = 1$ if there is an edge connecting nodes $i$ and $j$, and $A_{ij} = 0$ otherwise. In multi-slice cases, the spatial coordinates and gene expression profiles are denoted as $S = (S^{(1)}, S^{(2)}, \ldots, S^{(N_d)})$ and $X = (X^{(1)}, X^{(2)}, \ldots, X^{(N_d)})$, respectively, where $N_d$ represents the number of slices. SNN graphs $\mathcal{G} = (\mathcal{G}^{(1)}, \mathcal{G}^{(2)}, \ldots, \mathcal{G}^{(N_d)})$ are computed separately based on their corresponding spatial coordinates.

**Task Description.** Given the constructed SNN graph $\mathcal{G}$, our goal is to train a graph neural network (GNN) that generates spot representations without any label information, i.e., self-supervised learning. The trained GNN is then utilized for various downstream tasks, including spatial domain identification (SDI), multi-slice integration, and alignment.

## 4 METHODOLOGY

In this section, we introduce our method, Spotscape, which is a learning scheme for GNNs applied to the SRT data. In a nutshell, Spotscape learns spot representations by capturing global similarities between spots through the Similarity Telescope module (Sec 4.2), and refining them with cluster assignments using the prototypical contrastive module (Sec 4.3). Furthermore, Spotscape introduces the similarity scaling strategy (Sec 4.4) to balance intra- and inter-slice similarities, thereby alleviating batch effects. The overall framework of Spotscape is depicted in Figure 2.

### 4.1 MODEL ARCHITECTURE

In this work, we propose novel self-supervised learning strategies specifically tailored for SRT data, while adhering to a basic siamese network structure for our model architecture. In siamese network, we generate two augmented views, $\tilde{\mathcal{G}} = (\tilde{X}, \tilde{A})$ and $\tilde{\mathcal{G}}' = (\tilde{X}', \tilde{A}')$, by applying a stochastic graph augmentation $\mathcal{T}$ to the original graph $\mathcal{G}$, which consists of node feature masking and edge masking. Then, Spotscape computes spot representations $\tilde{Z} = f_\theta(\tilde{X}, \tilde{A})$ and $\tilde{Z}' = f_\theta(\tilde{X}', \tilde{A}')$, $f_\theta$ is a shared GNN-based encoder, $\tilde{Z} \in \mathbb{R}^{N_s \times D}$ and $\tilde{Z}' \in \mathbb{R}^{N_s \times D}$ represent spot representations derived from augmented graph $\tilde{\mathcal{G}}$ and $\tilde{\mathcal{G}}'$, respectively, and $D$ denotes the dimension size of representations.

## 4.2 SIMILARITY TELESCOPE WITH RELATION CONSISTENCY

Biological systems exhibit a continuous nature, where gene expression values vary smoothly along spatial coordinates. This continuity leads to feature similarities between neighboring spots, influenced both by their spatial proximity and functional characteristics. Therefore, relying solely on spatially neighboring spots provides limited information, highlighting the importance of reflecting the global context in this domain. While contrastive learning has become a standard for learning representations in the global context, it encounters limitations when applied to SRT data. This is primarily because the characteristics of individual cells cannot be fully defined individually, but they are influenced by the properties of neighboring cells within the tissue context. To address this, we propose a novel relation consistency loss for spot representation learning, which aims to capture the relationship between cells in the biological systems by reflecting the global context among multiple spots.

Specifically, given spot representations $\tilde{Z}$ and $\tilde{Z}'$, we propose to learn the consistent relationship that are invariant under augmentation as follows:

$$\mathcal{L}_{\text{SC}}(\tilde{Z}, \tilde{Z}') = \text{MSE}(\tilde{Z}_{norm} \cdot (\tilde{Z}'_{norm})^T, \tilde{Z}'_{norm} \cdot (\tilde{Z}_{norm})^T) \quad (1)$$

where $\tilde{Z}_{norm} \in \mathbb{R}^{N_s \times D}$ denotes the L2-normalized version of $\tilde{Z}$, and $MSE$ represents the Mean Squared Error. That is, we aim to minimize the cosine similarity between the spot representations that are obtained through differently augmented SNN graph. By doing so, the model learns consistent relationships, which is represented as cosine similarity, between all paired spots under different augmentations, capturing the continuous variations of spot representations across the entire slice.

Additionally, instead of relying on any predictor or stop gradient techniques (Thakoor et al., 2021) to avoid degenerate solutions, Spotscape simplifies the training procedure by employing a reconstruction loss as follows:

$$\mathcal{L}_{\text{Recon}}(X, \hat{X}, \hat{X}') = \text{MSE}(X, \hat{X}) + \text{MSE}(X, \hat{X}') \quad (2)$$

where $\hat{X} = g_\theta(\tilde{Z})$ and $\hat{X}' = g_\theta(\tilde{Z}')$ are reconstructed feature matrices predicted by a shared MLP decoder $g_\theta$ from each augmented view.

## 4.3 PROTOTYPICAL CONTRASTIVE LEARNING

While learning spot representations through the similarity telescope module, it is essential for these representations to be more fine-grained to enable more challenging downstream analyses, such as identifying rare cell types. To this end, Spotscape employs a prototypical contrastive learning scheme (Li et al., 2020a; De Donno et al., 2023; Lee et al., 2023) that groups semantically similar representations together while distancing dissimilar ones. Specifically, we obtain prototypes (i.e., centroids) by performing $K$-means clustering on spot representations $\tilde{Z}'$ derived from an augmented view $\tilde{\mathcal{G}}'$. Pairs of spots assigned to the same prototype are categorized as positive pairs, while pairs belonging to different prototypes are treated as negative pairs. This clustering process is repeated $T$ times with varying values of $K$ to identify semantically similar groups across different granularities. It is formally represented as follows:

$$l_{\text{PCL}}(\tilde{Z}_i, P_{\text{set}}) = \frac{1}{T} \sum_{t=1}^{T} \log \frac{e^{(\text{sim}(\tilde{Z}_i, p^t_{map_t(i)})/\tau)}}{\sum_{j=1}^{K_t} e^{(\text{sim}(\tilde{Z}_i, p^t_j)/\tau)}}, \quad (3)$$

where $\tau$ represents temperature, and $K_t$ indicates the number of clusters at each level of granularity during the $t$-th clustering iteration. $P_{\text{set}} = (P^1, ..., P^t, ..., P^T)$ represents the collection of prototype sets, with each $P^t = (p^t_1, p^t_2, ..., p^t_{k_t})$ containing the set of prototype representations for a specific granularity $t$. Additionally, $map_t(\cdot)$ denotes the mapping function that assigns each spot to a corresponding prototype based on the clustering assignments. By applying this to all spot representations, the overall prototypical contrastive learning (PCL) loss is given as follows:

$$\mathcal{L}_{\text{PCL}} = -\frac{1}{N_s} \sum_{i=1}^{N_s} l_{\text{PCL}}(\tilde{Z}_i, P_{\text{set}}). \quad (4)$$

Combining all of these losses, the final training loss for single-slice representation learning is formally defined as:

$$\mathcal{L}_{\text{Single}} = \lambda_{\text{SC}} \cdot \mathcal{L}_{\text{SC}} + \lambda_{\text{Recon}} \cdot \mathcal{L}_{\text{Recon}} + \lambda_{\text{PCL}} \cdot \mathcal{L}_{\text{PCL}} \quad (5)$$

Note to avoid the risk of obtaining inaccurate prototypes, the prototypical loss $\mathcal{L}_{\text{PCL}}$ gets involved in the training procedure after a warm-up period (500 epochs) of optimizing only the first two terms in Equation 8.

### 4.4 SIMILARITY SCALING STRATEGY

Beyond the single-slice SRT, multi-slice SRT allows for the analysis of gene expression patterns across multiple tissue sections. This provides a more comprehensive understanding of the spatial distribution and continuity of gene expression in entire tissues or organs, which could not have been achieved by the single-slice SRT. However, another challenge of learning representations from these multiple slices is the *batch effect*, where spot representations from the same slice are unexpectedly clustered together regardless of their biological significance, hindering researchers from obtaining useful representations related to biological functions. To alleviate this issue, given the SNN graph $\mathcal{G}^{(c)}$ and $\mathcal{G}^{(j)}$ of the current slice $c$ and another slice $j$, respectively, we explicitly regulate the scale of these similarities to maintain consistency across spots, as described below:

$$l_{\text{SS}}(H_i, \mathcal{G}^{(j)}) = \mathsf{Mean}_{s \in S_{\text{top}}^{(c)}}(H_i[s]) - \mathsf{Mean}_{s \in S_{\text{top}}^{(j)}}(H_i[s]), \quad \text{for} \quad i \in \mathcal{G}^{(c)}$$

$$\text{where} \quad S_{\text{top}}^{(c)} = \mathsf{Top\text{-}}k_{l \in \mathcal{G}^{(c)}}(H_i[l]) = (a_1, a_2, \ldots, a_k), \tag{6}$$

$$S_{\text{top}}^{(j)} = \mathsf{Top\text{-}}k_{l \in \mathcal{G}^{(j)}}(H_i[l]) = (b_1, b_2, \ldots, b_k)$$

Here, $H = \tilde{Z}_{norm}(\tilde{Z}'_{norm})^T \in \mathbb{R}^{N_s \times N_s}$ represents the similarity matrix that we optimize in the Similarity Telescope module, and $H_i[s]$ refers to the element in the $i$-th row and $s$-th column of this matrix. The set $S_{\text{top}}^{(c)}$ includes the top-$k$ most similar spots within the same slice as spot $i$, and the set $S_{\text{top}}^{(j)}$ includes the top-$k$ most similar spots in slice $j$. By doing this, Spotscape ensures that the distances between the top-$k$ spots remain nearly the same, regardless of the slice they belong to, effectively mixing all spots from different slices within the latent space. By extending it to all spots and slices, the final similarity scaling loss is given as follows:

$$\mathcal{L}_{\text{SS}} = \frac{1}{N_s(N_d - 1)} \sum_{i=1}^{N_s} \sum_{j=1}^{N_d} \mathbb{1}(i \notin \mathcal{G}^{(j)}) \cdot l_{\text{SS}}(H_i, \mathcal{G}^{(j)}) \tag{7}$$

where $\mathbb{1}(i \notin \mathcal{G}^{(j)})$ is the indicator function that equals 1 if $i$ is not included in $\mathcal{G}^{(j)}$ and 0 otherwise. Finally, the overall loss for multi-slice SRT data is formally represented as:

$$\mathcal{L}_{\text{Multi}} = \lambda_{\text{SC}} \cdot \mathcal{L}_{\text{SC}} + \lambda_{\text{Recon}} \cdot \mathcal{L}_{\text{Recon}} + \lambda_{\text{PCL}} \cdot \mathcal{L}_{\text{PCL}} + \lambda_{\text{SS}} \cdot \mathcal{L}_{\text{SS}} \tag{8}$$

where $\lambda_{\text{SS}}$ is additional balancing parameters of similarity scaling loss.

## 5 EXPERIMENTS

### 5.1 EXPERIMENTAL SETUP

**Datasets.** We conduct a comprehensive evaluation of Spotscape across five datasets derived from different technologies. For **single-slice experiments**, we use the dorsolateral prefrontal cortex (**DLPFC**) dataset, which includes 3 patients, each with 4 slices (12 slices in total). Additionally, we assess the middle temporal gyrus (MTG) dataset, comprising slices from a control group and an Alzheimer's disease (AD) group, as well as the Mouse embryo dataset. Lastly, we utilize Non-small cell lung cancer (NSCLC) data. In **multi-slice experiments**, we integrate the four slices from the same patient in the **DLPFC** dataset for the homogeneous integration task, while analyzing the differences between the control and AD groups in the **MTG** dataset for heterogeneous integration. Lastly, we evaluate heterogeneous alignment using the **Mouse embryo** dataset, where slices from different developmental stages require alignment to track developmental progression, and the Breast Cancer dataset, which includes spots corresponding to cancer cell types. Further details about data statistics can be found in Table 8 of Appendix A.

**Compared methods.** To ensure a fair comparison, we carefully select baseline methods based on their relevance to specific tasks. For the single-slice SDI task, we compare Spotscape with five state-of-the arts methods, i.e., SEDR (Xu et al., 2024), STAGATE (Dong & Zhang, 2022),

Table 1: Single-slice spatial domain identification performance on DLPFC data.

| | DLPFC (Patient 1) | | | | | | | | | | | | | | | |
| | Slice 151673 | | | | Slice 151674 | | | | Slice 151675 | | | | Slice 151676 | | | |
| | Silhouette | ARI | NMI | CA | Silhouette | ARI | NMI | CA | Silhouette | ARI | NMI | CA | Silhouette | ARI | NMI | CA |
|---|---|---|---|---|---|---|---|---|---|---|---|---|---|---|---|---|
| SEDR | 0.24(0.03) | 0.36(0.08) | 0.49(0.08) | 0.55(0.06) | 0.21(0.04) | 0.37(0.08) | 0.48(0.07) | 0.51(0.07) | 0.19(0.04) | 0.33(0.06) | 0.45(0.05) | 0.51(0.03) | 0.21(0.03) | 0.29(0.03) | 0.41(0.04) | 0.47(0.02) |
| STAGATE | 0.18(0.02) | 0.37(0.04) | 0.55(0.03) | 0.52(0.04) | 0.17(0.01) | 0.34(0.03) | 0.50(0.02) | 0.51(0.03) | 0.18(0.06) | 0.33(0.03) | 0.5(0.03) | 0.48(0.03) | 0.16(0.00) | 0.33(0.00) | 0.47(0.01) | 0.52(0.01) |
| SpaCAE | 0.35(0.05) | 0.21(0.01) | 0.37(0.01) | 0.43(0.01) | 0.27(0.02) | 0.25(0.03) | 0.38(0.01) | 0.44(0.03) | 0.22(0.02) | 0.23(0.03) | 0.41(0.03) | 0.42(0.04) | 0.27(0.02) | 0.23(0.02) | 0.34(0.02) | 0.43(0.03) |
| SpaceFlow | 0.43(0.04) | 0.42(0.06) | 0.57(0.05) | 0.57(0.03) | 0.39(0.03) | 0.37(0.04) | 0.51(0.03) | 0.53(0.03) | 0.41(0.03) | 0.38(0.07) | 0.55(0.06) | 0.53(0.04) | 0.41(0.02) | 0.38(0.05) | 0.51(0.05) | 0.53(0.04) |
| GraphST | 0.29(0.01) | 0.20(0.02) | 0.34(0.03) | 0.41(0.02) | 0.25(0.01) | 0.27(0.03) | 0.41(0.01) | 0.46(0.01) | 0.31(0.01) | 0.22(0.02) | 0.34(0.01) | 0.40(0.02) | 0.26(0.01) | 0.26(0.05) | 0.40(0.05) | 0.45(0.04) |
| Spotscape | **0.46**(0.01) | **0.47**(0.01) | **0.62**(0.02) | **0.62**(0.03) | **0.50**(0.02) | **0.45**(0.03) | **0.58**(0.02) | **0.60**(0.01) | **0.50**(0.03) | **0.46**(0.04) | **0.61**(0.02) | **0.60**(0.01) | **0.49**(0.01) | **0.41**(0.04) | **0.57**(0.03) | **0.55**(0.03) |
| | DLPFC (Patient 2) | | | | | | | | | | | | | | | |
| | Slice 151507 | | | | Slice 151508 | | | | Slice 151509 | | | | Slice 151510 | | | |
| | Silhouette | ARI | NMI | CA | Silhouette | ARI | NMI | CA | Silhouette | ARI | NMI | CA | Silhouette | ARI | NMI | CA |
| SEDR | 0.10(0.02) | 0.29(0.06) | 0.39(0.07) | 0.45(0.06) | 0.07(0.02) | 0.21(0.02) | 0.31(0.02) | 0.39(0.02) | 0.10(0.02) | 0.37(0.04) | 0.47(0.04) | 0.51(0.05) | 0.08(0.02) | 0.31(0.05) | 0.44(0.04) | 0.47(0.04) |
| STAGATE | 0.13(0.00) | 0.41(0.01) | 0.53(0.01) | 0.59(0.00) | 0.14(0.00) | 0.32(0.01) | 0.49(0.00) | 0.54(0.01) | 0.15(0.01) | 0.41(0.02) | 0.57(0.02) | 0.61(0.04) | 0.13(0.01) | 0.32(0.03) | 0.50(0.02) | 0.50(0.02) |
| SpaCAE | 0.27(0.04) | 0.28(0.06) | 0.41(0.06) | 0.46(0.06) | 0.29(0.03) | 0.20(0.04) | 0.31(0.05) | 0.40(0.04) | 0.32(0.01) | 0.31(0.01) | 0.44(0.02) | 0.50(0.04) | 0.28(0.02) | 0.27(0.02) | 0.42(0.03) | 0.45(0.02) |
| SpaceFlow | 0.39(0.02) | 0.55(0.03) | 0.68(0.02) | 0.71(0.05) | 0.36(0.03) | 0.44(0.04) | 0.57(0.03) | 0.58(0.04) | 0.38(0.03) | 0.53(0.05) | 0.66(0.02) | 0.65(0.04) | 0.37(0.02) | 0.5(0.03) | 0.64(0.01) | 0.61(0.02) |
| GraphST | 0.24(0.01) | 0.31(0.01) | 0.45(0.01) | 0.50(0.01) | 0.29(0.01) | 0.34(0.01) | 0.45(0.02) | 0.53(0.02) | 0.26(0.01) | 0.35(0.01) | 0.51(0.01) | 0.55(0.02) | 0.26(0.01) | 0.3(0.02) | 0.47(0.01) | 0.49(0.03) |
| Spotscape | **0.46**(0.01) | **0.58**(0.02) | **0.70**(0.02) | **0.73**(0.06) | **0.43**(0.02) | **0.48**(0.04) | **0.63**(0.02) | **0.63**(0.03) | **0.44**(0.01) | **0.55**(0.05) | **0.68**(0.03) | **0.65**(0.04) | **0.43**(0.02) | **0.51**(0.03) | **0.67**(0.01) | **0.61**(0.03) |
| | DLPFC (Patient 3) | | | | | | | | | | | | | | | |
| | Slice 151669 | | | | Slice 151670 | | | | Slice 151671 | | | | Slice 151672 | | | |
| | Silhouette | ARI | NMI | CA | Silhouette | ARI | NMI | CA | Silhouette | ARI | NMI | CA | Silhouette | ARI | NMI | CA |
| SEDR | 0.16(0.05) | 0.24(0.07) | 0.40(0.07) | 0.48(0.06) | 0.14(0.02) | 0.24(0.06) | 0.39(0.05) | 0.48(0.05) | 0.22(0.04) | 0.37(0.10) | 0.50(0.09) | 0.59(0.07) | 0.21(0.04) | 0.49(0.09) | 0.58(0.06) | 0.66(0.07) |
| STAGATE | 0.19(0.05) | 0.29(0.05) | 0.45(0.07) | 0.52(0.04) | 0.14(0.00) | 0.20(0.01) | 0.38(0.01) | 0.44(0.01) | 0.17(0.02) | 0.40(0.07) | 0.49(0.03) | 0.63(0.06) | 0.18(0.05) | 0.38(0.02) | 0.51(0.04) | 0.54(0.01) |
| SpaCAE | 0.30(0.02) | 0.21(0.02) | 0.28(0.03) | 0.43(0.02) | 0.27(0.07) | 0.21(0.03) | 0.28(0.02) | 0.43(0.04) | 0.38(0.16) | 0.38(0.16) | 0.29(0.01) | 0.49(0.05) | 0.32(0.07) | 0.25(0.04) | 0.35(0.05) | 0.50(0.01) |
| SpaceFlow | 0.44(0.03) | 0.30(0.07) | 0.48(0.03) | 0.51(0.05) | 0.42(0.03) | 0.34(0.05) | 0.50(0.03) | 0.56(0.05) | 0.43(0.04) | 0.54(0.04) | 0.67(0.02) | 0.67(0.04) | 0.46(0.01) | 0.60(0.06) | 0.70(0.02) | 0.73(0.06) |
| GraphST | 0.25(0.01) | 0.17(0.04) | 0.26(0.04) | 0.43(0.02) | 0.38(0.01) | 0.14(0.01) | 0.23(0.00) | 0.37(0.01) | 0.28(0.01) | 0.30(0.05) | 0.38(0.03) | 0.54(0.03) | 0.31(0.02) | 0.23(0.02) | 0.32(0.02) | 0.49(0.01) |
| Spotscape | **0.54**(0.02) | **0.45**(0.02) | **0.57**(0.02) | **0.65**(0.02) | **0.48**(0.01) | **0.45**(0.03) | **0.55**(0.01) | **0.66**(0.02) | **0.52**(0.06) | **0.59**(0.12) | **0.69**(0.05) | **0.72**(0.11) | **0.56**(0.04) | **0.72**(0.05) | **0.72**(0.02) | **0.82**(0.04) |

Table 2: Single-slice spatial domain identification performance on MTG data.

| | MTG - Control Group | | | | MTG - AD Group | | | |
| | Silhouette | ARI | NMI | CA | Silhouette | ARI | NMI | CA |
|---|---|---|---|---|---|---|---|---|
| SEDR | 0.46(0.03) | 0.41(0.02) | 0.59(0.02) | 0.52(0.02) | 0.32(0.06) | 0.43(0.08) | 0.59(0.07) | 0.57(0.07) |
| STAGATE | 0.35(0.01) | 0.54(0.00) | 0.65(0.00) | 0.59(0.00) | 0.27(0.01) | 0.51(0.01) | 0.61(0.01) | 0.59(0.01) |
| SpaCAE | 0.53(0.01) | 0.37(0.01) | 0.52(0.00) | 0.44(0.01) | 0.35(0.06) | 0.22(0.01) | 0.4(0.01) | 0.40(0.01) |
| SpaceFlow | 0.46(0.03) | 0.66(0.03) | 0.74(0.01) | 0.70(0.03) | 0.40(0.02) | 0.54(0.01) | 0.71(0.00) | 0.65(0.01) |
| GraphST | 0.49(0.01) | 0.38(0.00) | 0.51(0.00) | 0.48(0.00) | 0.34(0.02) | 0.43(0.01) | 0.55(0.05) | 0.55(0.04) |
| Spotscape | **0.53**(0.00) | **0.73**(0.02) | **0.78**(0.01) | **0.75**(0.02) | **0.48**(0.01) | **0.68**(0.02) | **0.75**(0.01) | **0.77**(0.03) |

Table 3: Single-slice SDI performance on Mouse Embryo data.

| | Mouse Embryo | | | |
| | Silhouette | ARI | NMI | CA |
|---|---|---|---|---|
| SEDR | 0.21(0.00) | 0.32(0.02) | 0.56(0.01) | 0.42(0.02) |
| STAGATE | 0.21(0.00) | 0.36(0.01) | 0.60(0.01) | 0.47(0.01) |
| SpaCAE | 0.23(0.00) | 0.34(0.01) | 0.60(0.01) | 0.48(0.00) |
| SpaceFlow | 0.29(0.01) | 0.42(0.01) | 0.60(0.02) | 0.49(0.03) |
| GraphST | 0.24(0.01) | 0.34(0.01) | 0.59(0.02) | 0.45(0.01) |
| Spotscape | **0.31**(0.01) | **0.45**(0.01) | **0.64**(0.01) | **0.54**(0.01) |

Table 4: Single-slice SDI performance on NSCLC data.

| | NSCLC | | | |
| | Silhouette | ARI | NMI | CA |
|---|---|---|---|---|
| SEDR | **0.40**(0.02) | 0.44(0.06) | 0.46(0.06) | 0.70(0.08) |
| STAGATE | 0.23(0.04) | 0.35(0.05) | 0.41(0.04) | 0.64(0.02) |
| SpaCAE | 0.13(0.01) | 0.32(0.05) | 0.38(0.03) | 0.62(0.02) |
| SpaceFlow | 0.37(0.02) | 0.53(0.03) | 0.52(0.02) | **0.75**(0.01) |
| GraphST | 0.16(0.00) | 0.30(0.03) | 0.38(0.00) | 0.65(0.00) |
| Spotscape | 0.38(0.02) | **0.58**(0.02) | **0.57**(0.01) | 0.74(0.01) |

SpaCAE (Hu et al., 2024), SpaceFlow (Ren et al., 2022), and GraphST (Long et al., 2023). For homogeneous integration, we add two more methods, PASTE (Zeira et al., 2022) and STAligner (Zhou et al., 2023), making a total of seven methods. For heterogeneous tasks, we compare with GraphST and STAligner(Zhou et al., 2023), while for heterogeneous alignment, we compare with STAligner and SLAT (Xia et al., 2023), both specialized for alignment tasks. Further details about each method's adoptable application can be found in table 9 of Appendix B.

**Evaluation Protocol.** Since Spotscape and all other baseline methods focus on learning representations for each spot, we first obtain the representations from each method and then apply the same evaluation tools for the subsequent downstream tasks. For single-slice spatial domain identification, we use $K$-means clustering on all the obtained representations and evaluate the results using Silhouette score, Adjusted Rand Index (ARI), Normalized Mutual Information (NMI), and Clustering Accuracy (CA). For multi-slice integration, we report the same clustering performance metrics as in the single-slice experiments and additionally include batch correction evaluation metrics such as Silhouette Batch, iLISI, kBET, and Graph Connectivity to assess the effectiveness of batch effect correction. For alignment, we make the alignment using the 'spatial matching' function provided by SLAT (Xia et al., 2023) and evaluate the Label Transfer ARI (LTARI), which measures the agreement between the true labels and the labels assigned through the alignment process, providing an evaluation of the alignment quality. To ensure a fair comparison, we conducted a hyperparameter search for all baseline methods and Spotscape. Since the optimal hyperparameters for each baseline method may vary across datasets, we identified the best-performing hyperparameters based on the NMI using the first seed. Details of the selected parameters and the corresponding search space are provided in Supplementary Section E All experimental results are averaged over 10 runs with different seeds, and the means and standard deviations are reported for each experiment.

## 5.2 SINGLE-SLICE EXPERIMENTAL RESULTS

Experimental results on three different datasets are reported in Table 1, Table 2 and 3, which show the SDI performance on the DLPFC, MTG, and Mouse Embryo datasets, respectively. From these results, we have the following observations: **1)** Spotscape consistently outperforms in all 15 slices across three datasets in terms of Silhouette score, ARI, NMI, CA. We argue that this is because Spotscape not only explores information from spatially local neighbors, which provides limited insights due to the continuous nature of SRT data, but also leverages information within a global

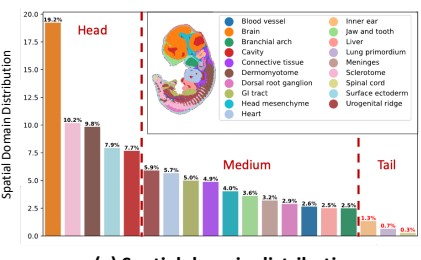 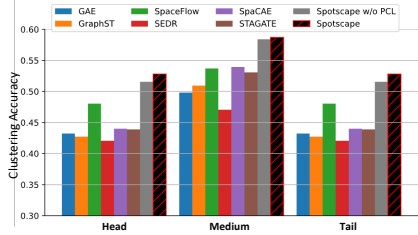

(a) Spatial domain distribution      (b) Head, Medium, Tail clustering performance comparison

Figure 3: (a) Spatial Domain distribution of Mouse Embryo data and (b) comparison of clustering accuracy across head, medium, and tail types.

context. **2)** Although previous methods like SpaceFlow, SpaCAE, and GraphST learn spot representations by incorporating the global context through Deep Infomax or contrastive learning and show generally better results than SEDR and STAGATE both of which only focus on the local view, optimizing similarities proves more beneficial for spatial domain identification, as it is closely related to the relative distance in the latent space. To further clarify this argument, we also conduct additional performance comparison with general self-supervised learning methods in Appendix B. **3)** Furthermore, to examine whether Spotscape effectively captures fine-grained information of rare cell types, we conduct a deeper analysis of the Mouse Embryo data, which displays imbalanced spatial domain distributions, as shown in Figure 3 (a). To achieve this, we initially categorize the cells into head, medium, and tail classes based on their distribution. The bottom 3 spatial domains, comprising less than 2%, were classified as the tail, while the top 5 domains showing significant changes in distribution were classified as the head, and the remaining domains were defined as medium, and then assess the performance for each class. As shown in Figure 3 (b), Spotscape outperforms the baselines across head, medium, and tail cell classes, highlighting its capability to capture fine-grained information of cells within rare spatial domains. Furthermore, we observe that model performance declines across all classes when the prototypical contrastive loss (Spotscape w/o PCL) is removed. This indicates that the prototypical contrastive loss enhances the model's ability to achieve fine-grained cell representation through a multi-granularity clustering approach, thereby contributing to clustering rare cell types.

### 5.3 MULTI-SLICE EXPERIMENTAL RESULTS

**Homogeneous Integration Results.** Among the multi-slice experiments, we first start with homogeneous integration tasks, which aim to integrate multiple slices from the homogeneous sample. To do so, we conduct experiments on the DLPFC data used for single-slice experiments, which consists of multiple slices obtained from vertical cuts of a single patient. Since these slices are from a single patient, they do not exhibit significant batch effects, enabling us to incorporate both multi-slice integration methods as well as single-slice SDI methods as baselines. As shown in Table 5, we observe Spotscape consistently outperforms all baseline methods, demonstrating its effectiveness in integrating information from the multiple slices from homogeneous sample.

**Heterogeneous Integration Results.** For the heterogeneous integration experiments, we assess the model's ability in integrating two distinct types of samples—the control group and the AD group in the MTG data—to analyze the differences between them. In this experiment, we also report batch effect correction metrics, such as Silhouette Batch, iLISI, kBET, and Graph Connectivity, to evaluate the effectiveness of correcting batch effects, along with clustering metrics. In Table 6, Spotscape demonstrates its effectiveness in integrating multi-slice data in terms of both clustering and batch effect correction, showing significantly better performance than the baselines. Moreover, in Figure 4, we observe that Spotscape's spot representations from different slices are well integrated while preserving their biological meaning. We also observe that the model performance significantly degrades without similarity scaling module (i.e., Spotscape w/o SS), while this effect is not as pronounced in homogeneous slices (Table 5), where batch effects are negligible. These results indicate the effectiveness of the similarity scaling module in mitigating batch effects when handling multiple slices.

Table 5: Homogeneous integration performance on DLPFC data.

| | Patient 1 | | | | Patient 2 | | | | Patient 3 | | | |
|---|---|---|---|---|---|---|---|---|---|---|---|---|
| | Silhouette | ARI | NMI | CA | Silhouette | ARI | NMI | CA | Silhouette | ARI | NMI | CA |
| SEDR | 0.30 (0.02) | 0.38 (0.06) | 0.49 (0.06) | 0.56 (0.06) | 0.22 (0.03) | 0.32 (0.05) | 0.44 (0.07) | 0.48 (0.07) | 0.31 (0.02) | 0.43 (0.02) | 0.51 (0.01) | 0.56 (0.03) |
| STAGATE | 0.16 (0.03) | 0.31 (0.03) | 0.46 (0.03) | 0.49 (0.03) | 0.10 (0.01) | 0.30 (0.02) | 0.46 (0.01) | 0.48 (0.02) | 0.14 (0.03) | 0.31 (0.09) | 0.43 (0.06) | 0.54 (0.08) |
| SpaCAE | 0.21 (0.01) | 0.21 (0.03) | 0.36 (0.02) | 0.40 (0.02) | 0.13 (0.03) | 0.12 (0.06) | 0.19 (0.07) | 0.32 (0.05) | 0.20 (0.05) | 0.13 (0.05) | 0.14 (0.05) | 0.43 (0.06) |
| SpaceFlow | 0.31 (0.01) | 0.48 (0.03) | 0.60 (0.02) | 0.60 (0.02) | 0.27 (0.02) | 0.44 (0.05) | 0.59 (0.02) | 0.58 (0.04) | 0.30 (0.03) | 0.51 (0.02) | 0.60 (0.01) | 0.69 (0.05) |
| GraphST | 0.30 (0.02) | 0.18 (0.01) | 0.32 (0.01) | 0.38 (0.02) | 0.30 (0.01) | 0.25 (0.01) | 0.39 (0.01) | 0.42 (0.02) | 0.30 (0.01) | 0.25 (0.04) | 0.30 (0.04) | 0.50 (0.01) |
| PASTE | 0.15 (0.00) | 0.34 (0.00) | 0.45 (0.00) | 0.54 (0.00) | 0.11 (0.00) | 0.17 (0.00) | 0.28 (0.00) | 0.40 (0.00) | 0.11 (0.00) | 0.29 (0.00) | 0.43 (0.00) | 0.54 (0.00) |
| STAligner | 0.34 (0.04) | 0.38 (0.04) | 0.52 (0.04) | 0.55 (0.04) | 0.20 (0.04) | 0.29 (0.02) | 0.45 (0.04) | 0.48 (0.03) | 0.24 (0.04) | 0.37 (0.05) | 0.47 (0.05) | 0.59 (0.06) |
| Spotscape (w/o SS) | 0.41 (0.01) | **0.56** (0.01) | **0.69** (0.01) | 0.67 (0.02) | 0.39 (0.01) | **0.53** (0.02) | 0.67 (0.01) | **0.69** (0.03) | 0.39 (0.02) | 0.58 (0.06) | **0.67** (0.02) | 0.75 (0.06) |
| Spotscape | **0.42** (0.01) | **0.56** (0.02) | **0.69** (0.01) | **0.68** (0.02) | **0.40** (0.02) | **0.53** (0.02) | **0.68** (0.01) | **0.69** (0.02) | **0.40** (0.02) | **0.60** (0.04) | **0.67** (0.01) | **0.76** (0.04) |

Table 6: Heterogeneous integration performance on MTG data.

| | Clustering Metric | | | | Batch Effect Correction Metric | | | |
|---|---|---|---|---|---|---|---|---|
| | Silhouette | ARI | NMI | CA | Silhouette Batch | iLISI | kBET | Graph Connectivity |
| GraphST | 0.43 (0.01) | 0.23 (0.02) | 0.42 (0.00) | 0.39 (0.01) | 0.56 (0.00) | 0.00 (0.00) | 0.02 (0.00) | 0.65 (0.02) |
| STAligner | 0.38 (0.03) | 0.38 (0.03) | 0.54 (0.03) | 0.49 (0.02) | 0.62 (0.04) | **0.16** (0.23) | 0.11 (0.08) | 0.85 (0.04) |
| Spotscape (w/o SS) | **0.59** (0.04) | 0.40 (0.07) | 0.56 (0.03) | 0.52 (0.05) | 0.25 (0.02) | 0.00 (0.00) | 0.00 (0.00) | 0.64 (0.01) |
| Spotscape | 0.52 (0.03) | **0.68** (0.08) | **0.75** (0.02) | **0.76** (0.08) | **0.69** (0.01) | 0.08 (0.04) | **0.17** (0.05) | **0.88** (0.02) |

Table 7: Alignment performance of Mouse embryo datasets.

| | LTARI |
|---|---|
| STAligner | 0.46 (0.02) |
| scSLAT | 0.52 (0.01) |
| Spotscape | **0.56** (0.01) |

Figure 4: UMAP of Raw, GraphST, STAligner, Spotscape (w/o SS), Spotscape by slice, ground truth, and $K$-means clustering results

Figure 5: Alignment results of Mouse embryo datasets.

To check whether our results yield biologically meaningful results, we investigate differentially expressed genes (DEGs) and their biological functions between the control and Alzheimer's disease (AD) group through Gene Ontology (GO) enrichment analysis for each cluster, representing a cortical layer in a brain. Since Spotscape provides spatially organized and reliably distributed clusters as actual cortical layers in a brain, all clusters are assigned to the cortical layers. As pathological influence of AD on different cortical layers is diverse, it is highly worthwhile to identify differences between the control and AD in each region (Romito-DiGiacomo et al., 2007). As depicted in Figure 24, in layer 2, which is regarded as a superficial layer, the terms in 'humoral immune response mediated by circulating immunoglobulin (GO:0002455)', 'synapse pruning (GO:0098883)', and 'regulation Of histone deacetylase activity (GO:1901725)' are enriched. On the other hand, layer 5, a deeper layer, enrich terms as 'synapse pruning (GO:0098883)', 'positive regulation of cytokine production (GO:0001819)', 'microglial cell activation (GO:0001774)', and 'Positive regulation of neuron death (GO:1901216)', as shown in Figure 25. All of these enriched biological processes are reported to be considerably relevant with AD (Mruthinti et al., 2004; Brucato & Benjamin, 2020; Lu et al., 2015; Wu et al., 2021; Goel et al., 2022). Moreover, the top enriched molecular function in Layer 5 is 'amyloid-beta binding (GO:0001540)', supporting reliability of results. Interestingly, synapse pruning and terms related to immune response are remarkably enriched in common, while angiogenesis, known to be associated with amyloid-beta pathway in AD (WA et al., 2013), is only enriched in Layer 2. These observations provide biological insights, namely shared characteristics and difference of AD in distinct cortical layers.

**Multi-slice Alignment Results.** Finally, we conduct experiments on multi-slice alignments of the Mouse Embryo data, which require alignment results to track the development stages of the embryo. To this end, we match E11.5 and E12.5 and report the Label Transfer ARI (LTARI) in Table 7, which measures the agreement between true labels and the labels assigned through the alignment process, and visualize our results in Figure 5. These results show that Spotscape achieves better alignment than SLAT, which is specifically designed for alignment tasks, demonstrating the general applicability of Spotscape.

Furthermore, we conduct cross-technology alignment between data obtained from Xenium and Visium. Since Xenium offers higher resolution than Visium, while Visium provides a more comprehensive transcriptome view, aligning Xenium with Visium creates a complementary approach that combines the strengths of both: high resolution and broader coverage. To this end, we align triple-positive cells in Xenium—those positively enriched for the ERBB2, PGR, and ESR1 marker genes associated with breast tumors—with corresponding Visium spots.

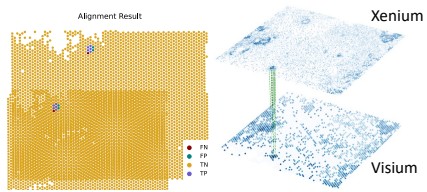

Figure 6: Alignment results of triple positive cells.

Figure 6 shows that Spotscape successfully outputs seven aligned points and identifies five triple-positive cells in the Visium data. This demonstrates the superiority of Spotscape, as it can successfully align extremely rare cell types (e.g., cancer cells).

### 5.4 MODEL ANALYSIS

**Ablation studies.** We also conduct ablation studies on the components of Spotscape to clarify the necessity of each module, as shown in Figure 7. Across all three tasks, our proposed Similarity Telescope (i.e., $\mathcal{L}_{SC}$) demonstrates its importance by showing a significant performance drop without this module. Additionally, prototypical contrastive learning (i.e., $\mathcal{L}_{PCL}$) further confirms its role in enhancing representations by consistently showing performance gains. In contrast, the reconstruction loss (i.e., $\mathcal{L}_{Recon}$) does not demonstrate significant performance gains excluding alignment tasks, since it is only needed for stabilizing the training procedures.

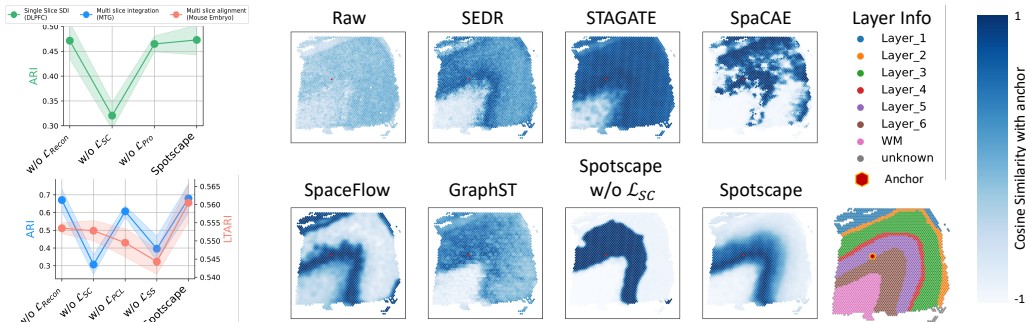

Figure 7: Ablation studies.          Figure 8: Similarity comparison based on anchor node 489 in Layer 5.

**Similarity analysis.** As a deeper analysis of Spotscape, we examine whether it successfully learns the relative similarities between spots, which is a key motivation behind our approach. In Figure 8, we randomly select an anchor spot from the DLPFC data and visualize the similarity between the selected anchor and other remaining spots. While other baselines fail to capture appropriate similarities, Spotscape accurately reflects the dynamics of the SRT data with respect to the spatial distance and exhibits varying levels of similarity corresponding to true spatial domain types.

### 6 CONCLUSION

In this work, we propose Spotscape, a novel framework for representation learning on the SRT data that is generally adaptable for both single and multi-slice tasks. The main idea of Spotscape is that while the spatial locality information is important in the SRT data, it often provides limited insights due to the continuous nature of this data. Therefore, Spotscape reflects the global similarities between spot representations by preserving a global similarity map invariant to augmentations during the training process. Moreover, Spotscape enhances spot representations by introducing the prototypical contrastive learning scheme into the SRT data to learn more fine-grained spot representations. Furthermore, we introduce a simple batch effect reduction strategy called similarity scaling, which explicitly regulates the scale of similarities to maintain consistency across spots located in different samples for extending applications of Spotscape to multi-slice tasks. Extensive experiments demonstrate that Spotscape outperforms existing baselines across SRT data from various platforms and diverse downstream tasks. Furthermore, we show that results from Spotscape can assist biologically meaningful findings, highlighting its future potential for practical SRT analysis.

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

# Supplementary Material

*- Global Context-aware Representation Learning for Spatially Resolved Transcriptomics -*

# A  DATASETS

Table 8: Statistics for datasets used for experiments.

| Data | Species | Tissue | Technology | Resolution | Cells/Spots | Genes | # of Spatial Domains | Reference |
|------|---------|--------|-----------|-----------|-------------|-------|---------------------|-----------|
| DLPFC | Human | Brain (dorsolateral prefrontal cortex; DLPFC) | 10x Visium | 50 $\mu$m | 3460 ~ 4789 | 33538 | 5 ~ 7 | (Maynard et al., 2021) |
| MTG | Human | Brain (middle temporal gyrus; MTG) | 10x Visium | 50 $\mu$m | 3445 ~ 4832 | 36601 | 6 ~ 7 | (Chen et al., 2022b) |
| Mouse Embryo | Mouse | Whole embryo | Stereo-seq | 0.2 $\mu$m | 30756 ~ 55295 | 25485 ~ 27330 | 18~19 | (Chen et al., 2022a) |
| NSCLC | Human | Non-small cell lung cancer (NSCLC) | CosMX | Subcellular | 960 | 11756 | 4 | (Bhuva et al., 2024) |
| Breast Cancer | Human | Breast Cancer | 10x Visium | 50 $\mu$m | 4992 | 18085 | 11 | (Janesick et al., 2023) |
| Breast Cancer | Human | Breast Cancer | 10x Xenium | Subcellular | 167780 | 313 | 20 | (Janesick et al., 2023) |

In this section, we compare Spotscape with baseline methods on various datasets. The data statistics are in Table 8.

**Human Dorsolateral Prefrontal Cortex (DLPFC).** It comprises 12 tissue slices from 3 adult samples, with 4 consecutive slices per sample, derived from the dorsolateral prefrontal cortex. These slices were profiled using the 10x Visium platform. The original study manually annotated 6 neocortical layers (layers 1 to 6) as well as the white matter (see Figure 9).

**Middle Temporal Gyrus (MTG).** The MTG (middle temporal gyrus) dataset includes samples from both control and Alzheimer's disease (AD) groups. The MTG is a brain region particularly vulnerable to early AD pathology. In the original study, spatial transcriptomics profiles were characterized for both AD and control MTG samples by the 6 neocortical layers (layer 1 to 6) and white matter, utilizing the 10x Visium platform for detailed tissue profiling. The spot distribution is denoted in Figure 10.

**Mouse Embryo.** It is mouse whole embryo datasets by development stages. It was profiled by Stereo-seq technology, which allows spatial transcriptomics at the cellular level by integrating DNA nanoball-patterned arrays with in situ RNA capture. It offers a detailed spatiotemporal transcriptomic atlas (MOSTA) of mouse embryonic development (see Figure 12).

**Non-small cell lung cancer (NSCLC).** The dataset comprises high-resolution, subcellular-level spatial transcriptomics data from human lung tissue, encompassing four distinct spatial domains (see Figure 11), including a tumor region. This data was generated using the NanoString CosMX platform.

**Human Breast Cancer**. It comprises spatial transcriptomics of human breast cancer tissues using 10x Visium for whole-transcriptome spatial data and 10x Xenium for high-resolution gene expression at the subcellular level. This combined approach offers detailed mapping of tumor microenvironments (see Figure 6), highlighting molecular differences and cell-type composition to better understand cancer heterogeneity and invasion.

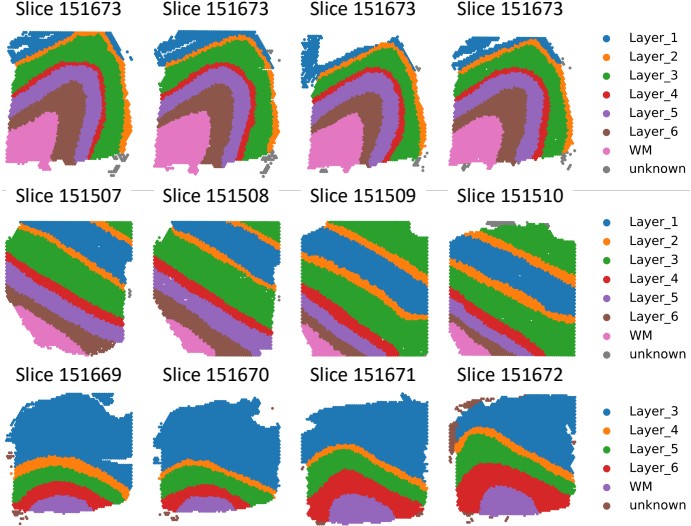

Figure 9: Spatial coordinates of DLPFC dataset.

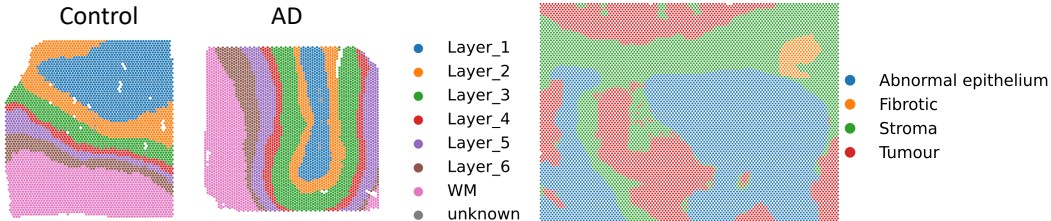

Figure 10: Spatial coordinates of MTG dataset. Figure 11: Spatial coordinates of NSCLC dataset.

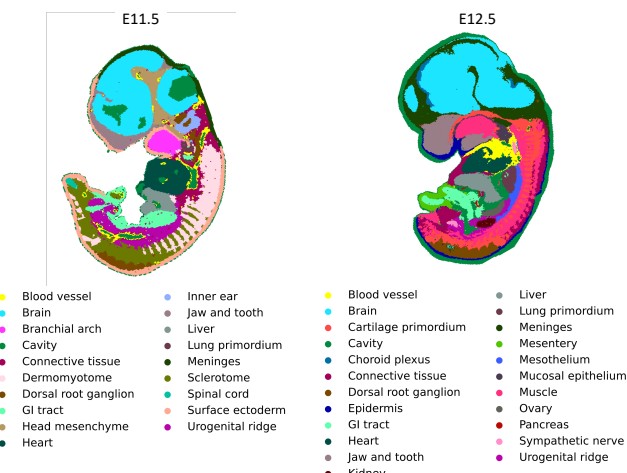

Figure 12: Spatial coordinates of Mouse Development dataset.

# B  BASELINE METHODS

In Table 9, we indicate which baseline methods are applicable to specific tasks, categorizing them based on whether their respective papers address those problems. Furthermore, we compare the performance of Spotscape with general self-supervised representation learning schemes. Graph Contrastive Learning (Chen et al., 2020; Zhu et al., 2020) is a instance-wise contrastive learning method that learns representations by pushing negative pairs apart and pulling positive pairs together. BGRL (Thakoor et al., 2021; Grill et al., 2020) is a consistency regularization method that learns representations by enforcing consistency between two differently augmented views. SwAV (Caron et al., 2020b) learns representations by minimizing the difference between two cluster assignments that are obtained through optimal transport. Barlow twins (Caron et al., 2020a) learns representations by minimizing redundancy between two augmented view. Although these methods demonstrate strong performance across various domains, our results in Figure 13 indicate that Spotscape is the most suitable model for SRT data, emphasizing its effectiveness in this context.

Table 9: Baseline methods and their application across various tasks

| Method | Single-slice SDI | Homogeneous integration | Homogeneous alignment | Heterogeneous integration | Heterogeneous alignment |
|---|---|---|---|---|---|
| SEDR | ✓ | | | | |
| STAGATE | ✓ | | | | |
| SpaCAE | ✓ | | | | |
| SpaceFlow | ✓ | | | | |
| GraphST | ✓ | ✓ | | | |
| PASTE | | ✓ | ✓ | | |
| STAligner | | ✓ | ✓ | ✓ | ✓ |
| SLAT | | | ✓ | | ✓ |
| Spotscape | ✓ | ✓ | ✓ | ✓ | ✓ |

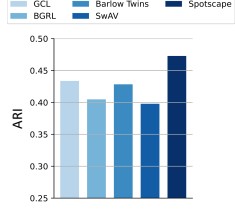

Figure 13: Comparison with self-supervised learning

## C    PSEUDO CODE

In this section, we provide pseudocode of Spotscape in Algorithm 1.

---

**Algorithm 1** Overall framework of Spotscape

**Require:** Spatial nearest neighbor graph $\mathcal{G} = (X, A)$, feature matrix $X$, adjacency matrix $A$, graph augmentation $\mathcal{T}$, GCN encoder $f_\theta$, decoder $g_\theta$, number of slices $N_d$, number of spots $N_s$, number of latent dimensions $D$, loss balancing parameters $(\lambda_{Recon}, \lambda_{SC}, \lambda_{PCL}, \lambda_{SS})$, temperature $\tau$, learning rate $\eta$

**Ensure:** Node embeddings $Z$, reconstructed feature matrix $\hat{X}$

1: **for** epoch **in** epochs:
2:      $\tilde{\mathcal{G}}, \tilde{\mathcal{G}}' = \mathcal{T}(\mathcal{G})$                                    /* two randomly augmented version of G */

3:      **Step 1: Graph Autoencoder**
4:      $\tilde{Z} = f_\theta(\mathcal{G}), \ \tilde{Z}' = f_\theta(\tilde{\mathcal{G}}')$                     /* compute spot embedding using GNN encoder */
5:      $\hat{X} = g_\theta(\tilde{Z}), \ \hat{X}' = g_\theta(\tilde{Z}')$                    /* reconstruct the feature matrix using decoder */

6:      **Step 2: Similarity Telescope with Relation Consistency** (Section 4.2)
7:      $\mathcal{L}_{Recon} =$ Reconstruction Loss$(X, \hat{X}, \hat{X}')$                                 (Eqn. 2)
8:      $\mathcal{L}_{SC}, H =$ Similarity Telescope with Relation Consistency Loss$(\tilde{Z}, \tilde{Z}')$

9:      **Step 3: Prototypical Contrastive Learning** (Section 4.3)
10:     **if** epoch $\geq$ warm-up epoch **then**
11:         $\mathcal{L}_{PCL} =$ PCL Loss$(\tilde{Z}, \tilde{Z}')$
12:     **else**
13:         $\mathcal{L}_{PCL} = 0$
14:     **end if**

15:     **Step 4: Similarity Scaling Strategy** (Section 4.4)
16:     **if** $N_d \geq 2$ **then**
17:         $\mathcal{L}_{SS} =$ Similarity Scaling Loss$(H, \mathcal{G})$                                 (Eqn. 7)
18:     **else**
19:         $\mathcal{L}_{SS} = 0$
20:     **end if**

21:     **Step 5: Compute Loss**
22:     $\mathcal{L} = \lambda_{Recon}\mathcal{L}_{Recon} + \lambda_{SC}\mathcal{L}_{SC} + \lambda_{PCL}\mathcal{L}_{PCL} + \lambda_{SS}\mathcal{L}_{SS}$

23:     **Step 6: Backpropagation and Parameter Update**
24:     Update parameters $\theta$ using Adam optimizer: $\theta_{epoch} \leftarrow$ Adam$(\theta_{epoch-1}, \eta)$

25: **Return:** Node embeddings $Z$, reconstructed feature matrix $\hat{X}$

     /* Utility Functions */
26: **Function** Similarity Telescope with Relation Consistency Loss$(\tilde{Z}, \tilde{Z}')$:
27:     $\tilde{Z}_{norm} =$ L2-norm$(\tilde{Z}), \ \tilde{Z}'_{norm} =$ L2-norm$(\tilde{Z}')$              /* L2-normalization */
28:     $H = \tilde{Z}_{norm} \cdot (\tilde{Z}'_{norm})^T, \ H' = \tilde{Z}'_{norm} \cdot (\tilde{Z}_{norm})^T$         /* compute cosine similarity */
29:     $\mathcal{L}_{SC} =$ MSE$(H, \ H')$                                               (Eqn. 1)
30:     **Return:** $\mathcal{L}_{SC}, H$

31: **Function** PCL Loss$(\tilde{Z}, \tilde{Z}')$:
32:     # $P_{set}$: the collection of prototype sets from K-means clustering
33:     $P_{set} \leftarrow$ Assign Prototype$(\tilde{Z}')$
34:     Calculate the prototypical contrastive loss $\mathcal{L}_{PCL}$ using $\tilde{Z}$ and $P_{set}$         (Eqn. 4)
35:     **Return:** $\mathcal{L}_{PCL}$

36: **Function** Assign Prototype$(Z)$:
37:     $P_{set} \leftarrow [\,]$
38:     **for** $K$ **in** $[K^1, K^2, \ldots, K^T]$:
39:         Cluster each cell into $K$ clusters based on $Z$
40:         Compute a prototype matrix $P \in \mathbb{R}^{K \times D}$ by averaging of the spot embeddings per cluster
41:         Append $P$ to $P_{set}$
42:     **Return:** $P_{set}$

---

## D SENSITIVITY ANALYSIS

We conduct a sensitivity analysis on all four balancing parameters $\lambda_{Recon}$, $\lambda_{SC}$, $\lambda_{PCL}$, and $\lambda_{SS}$ in Figure 14, 15, 16, and 17, respectively. In the case of the reconstruction loss ($\lambda_{Recon}$), when its weight is too high, performance tends to degrade, indicating that it serves primarily as an auxiliary loss to prevent degenerate solutions. On the other hand, the relation consistency loss ($\lambda_{SC}$) shows a degradation in performance when its weight is too small, emphasizing the importance of reflecting global similarities through this loss in Spotscape. Prototypical contrastive learning ($\lambda_{PCL}$) is robust within a reasonable search space and does not dominate the overall training process. However, it leads to significant performance drops when its weight is too high. Finally, similarity scaling ($\lambda_{SS}$) shows robust performance across a wide range of values, with slightly improved performance at higher weights. Furthermore, we conduct a sensitivity analysis for the manually tuned parameters, namely $\tau$ and the learning rate, as shown in Figures 18, and 19. We observe that $\tau$ shows generally robust performance, while the learning rate fluctuates significantly without a clear trend. These results provide insight that, except for the learning rate, other hyperparameters exhibit robustness within a reasonable search space, suggesting that Spotscape requires some learning rate search strategies.

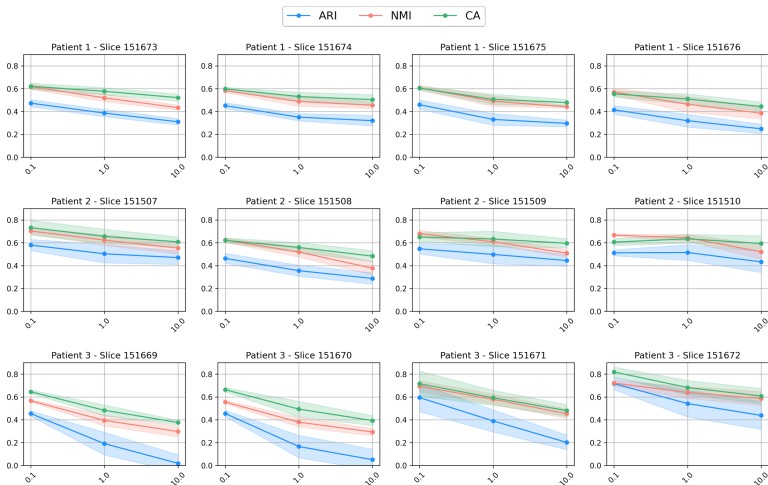

Figure 14: Sensitivity analysis for reconstruction loss balancing parameter ($\lambda_{Recon}$) of single DLPFC.

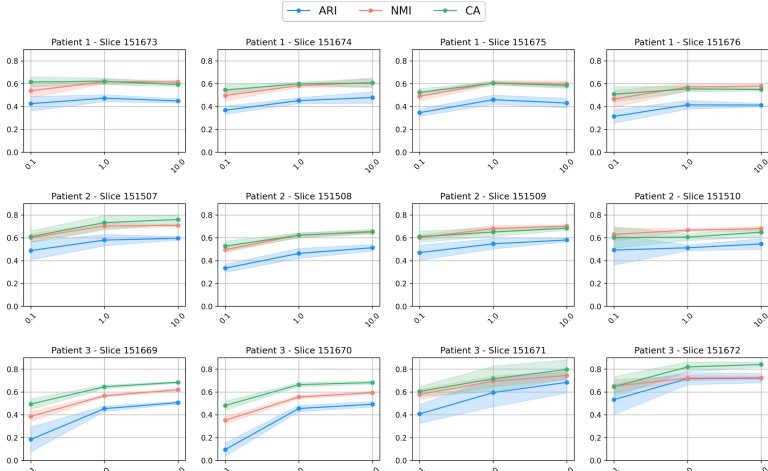

Figure 15: Sensitivity analysis for similarity telescope loss balancing parameter ($\lambda_{SC}$) of single DLPFC.

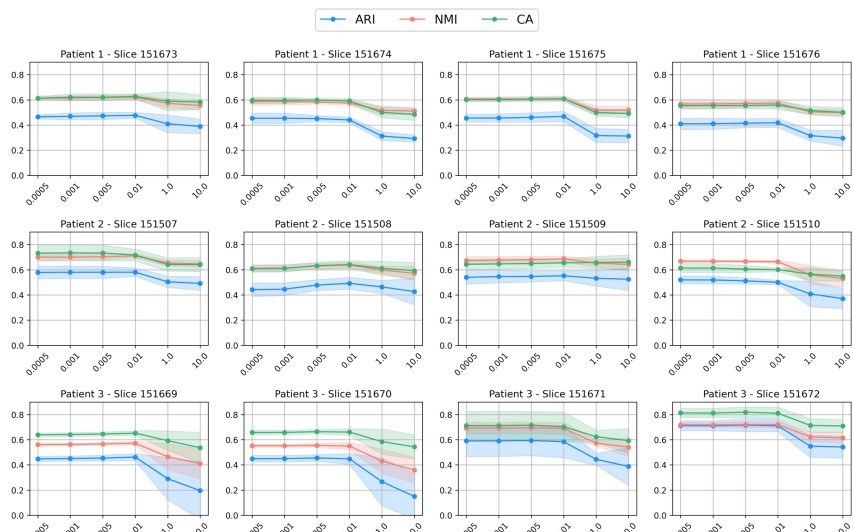

Figure 16: Sensitivity analysis for balancing parameter of PCL ($\lambda_{PCL}$) of single DLPFC.

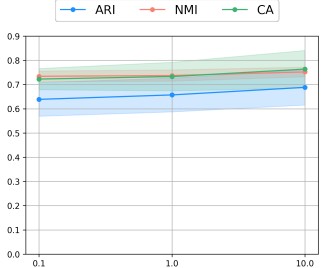

Figure 17: Sensitivity analysis for balancing parameter of similarity scaling ($\lambda_{SS}$) of MTG.

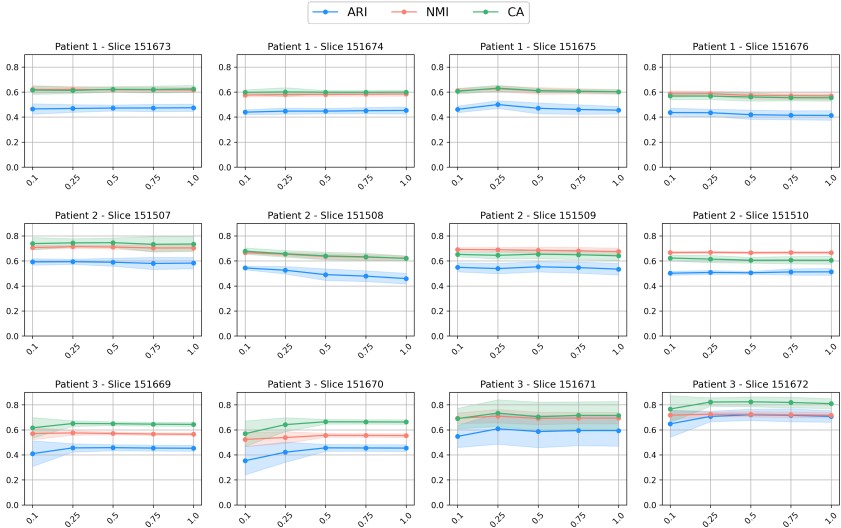

Figure 18: Sensitivity analysis for tau ($\tau$) of single DLPFC.

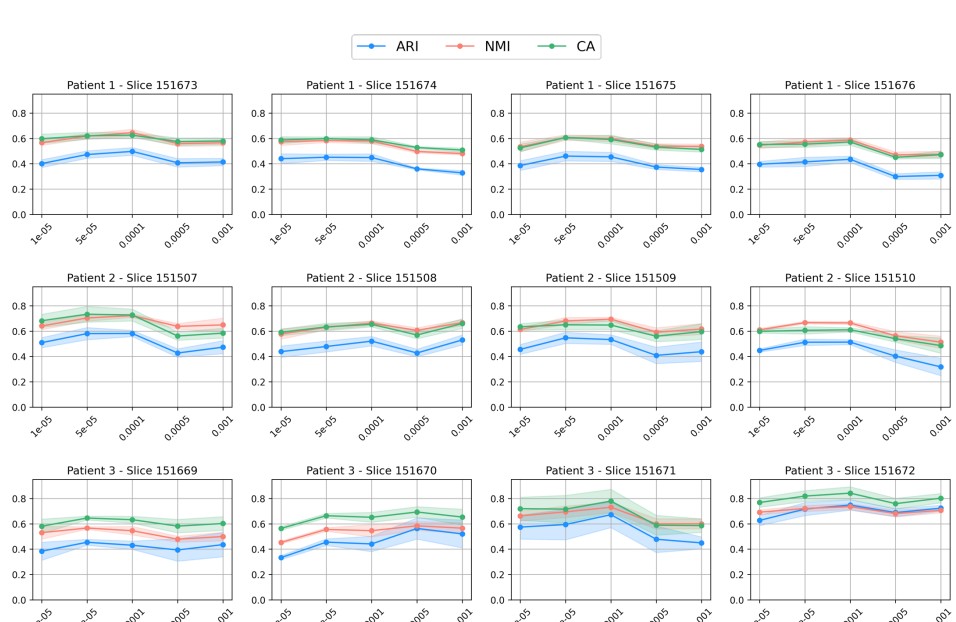

Figure 19: Sensitivity analysis for learning rate of single DLPFC.

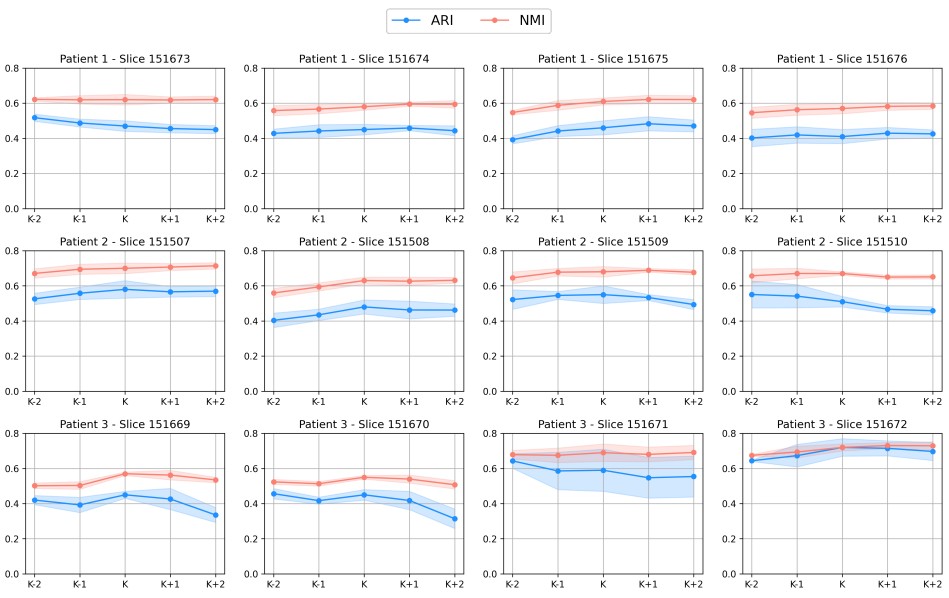

Figure 20: Sensitivity analysis for number of cluster (K) of single DLPFC.

# E  HYPERPARAMETER SELECTION AND IMPLEMENTATION DETAILS

## E.1  HYPERPARAMTER SEARCH FOR MODEL PERFORMANCE COMPARISON

To ensure a fair comparison, we conducted a hyperparameter search for both Spotscape and the baseline methods. The best-performing hyperparameters were selected by evaluating the NMI with the first seed. Specifically, for Spotscape, the hyperparameter search spaces were defined as follows: for $\lambda_{PCL}$, the values considered were $\{0.0005, 0.001, 0.005, 0.01\}$; for $\lambda_{SS}$, the range included $\{0.1, 1.0, 10.0\}$. The temperature ($\tau$) in PCL was explored over $\{0.1, 0.25, 0.5, 0.75, 1.0\}$, and the learning rate search space consisted of $\{0.00001, 0.00005, 0.0001, 0.0005, 0.001\}$. The remaining hyperparameters were fixed, and the ones used to report the experimental results are listed in Table 10.

Table 10: Hyperparameter settings of Spotscape

| | Fixed | DLPFC Single | MTG Single | Mouse Embryo | DLPFC Multi Integration | MTG Multi Integration | Mouse Embryo Alignment | Visium - Xenium Alignment |
|---|---|---|---|---|---|---|---|---|
| $\lambda_{Recon}$ | ✔ | 0.1 | 0.1 | 0.1 | 0.1 | 0.1 | 0.1 | 0.1 |
| $\lambda_{SC}$ | ✔ | 1.0 | 1.0 | 1.0 | 1.0 | 1.0 | 1.0 | 1.0 |
| $\lambda_{PCL}$ | | 0.005 | 0.0005 | 0.0005 | 0.005 | 0.01 | 0.01 | 0.01 |
| $\lambda_{SS}$ | | N/A | N/A | N/A | 0.1 | 10.0 | 1.0 | 1.0 |
| GCN encoder dimensions | ✔ | [256, 64] | [256, 64] | [256, 64] | [256, 64] | [256, 64] | [256, 64] | [256, 64] |
| $\tau$ | | 0.75 | 1.0 | 0.1 | 0.5 | 0.5 | 0.5 | 0.5 |
| Top-$k$ | ✔ | 5 | 5 | 5 | 5 | 5 | 5 | 5 |
| Training epochs | ✔ | 1000 | 1000 | 1000 | 1000 | 1000 | 1000 | 1000 |
| Warm-up epochs | ✔ | 500 | 500 | 500 | 500 | 500 | 500 | 500 |
| Learning rate | | 0.00005 | 0.0001 | 0.00001 | 0.0005 | 0.001 | 0.00001 | 0.00001 |
| Feature masking rate ($\mathcal{T}_{f,1}$) | ✔ | 0.2 | 0.2 | 0.2 | 0.2 | 0.2 | 0.2 | 0.2 |
| Feature masking rate ($\mathcal{T}_{f,2}$) | ✔ | 0.2 | 0.2 | 0.2 | 0.2 | 0.2 | 0.2 | 0.2 |
| Edge masking rate ($\mathcal{T}_{e,1}$) | ✔ | 0.2 | 0.2 | 0.2 | 0.2 | 0.2 | 0.2 | 0.2 |
| Edge masking rate ($\mathcal{T}_{e,2}$) | ✔ | 0.2 | 0.2 | 0.2 | 0.2 | 0.2 | 0.2 | 0.2 |

Additionally, we conducted a grid search primarily targeting the learning rate and loss balancing parameters for the baseline models. The learning rates for all baselines were explored within the search space $\{0.00001, 0.00005, 0.0001, 0.0005, 0.001, 0.005, 0.01, 0.05\}$. Similarly, the loss balancing parameters were tuned across the range $\{0.1, 1.0, 10.0\}$ including their default parameter. More precisely, for SEDR, it searched learning rate and balance parameters regarding reconstruction loss, VGAE loss, and self-supervised loss. For STAGATE, the search focused solely on the learning rate. In the case of SpaCAE, both the learning rate and the spatial expression augmentation parameter ($\alpha$) were tuned within $\{0.5, 1.0\}$. SpaceFlow was optimized by adjusting the learning rate and the spatial consistency loss balancing parameter. For GraphST, we explored the learning rate and the balancing parameters for feature reconstruction loss and self-supervised contrastive loss. Regarding STAligner, we searched for the optimal learning rates for both the pretrained model (i.e., STAGATE) and the fine-tuning process. Finally, for scSLAT, we applied the default parameters since the experiments were conducted under identical settings and with the same dataset. This systematic parameter-tuning process facilitated the effective optimization of each baseline model's performance.

## E.2  UNSUPERVISED HYPERPARAMETER SEARCH STRATEGY

To apply Spotscape to new data, an appropriate hyperparameter search strategy is essential. Fortunately, Spotscape is largely robust to hyperparameters, with the exception of the learning rate, which is inherently sensitive in gradient-based optimization models. For this reason, we fix all parameters except the learning rate and search for the learning rate that maximizes the silhouette score, which can be achieved without any supervised information. Specifically, $\lambda_{PCL}$, $\lambda_{SS}$, and $\tau$ are set to 0.0005, 10, and 0.75, respectively, while the learning rate is selected from the set $\{0.00001, 0.00005, 0.0001, 0.0005, 0.001\}$. Using this hyperparameter optimization strategy, we obtained the hyperparameters listed in Table 11 and reported the computed silhouette scores during the search process for DLPFC in Figure 21. We then compared the performance of the hyperparameters optimized without supervision with that of the hyperparameters optimized with supervision, which were used solely for performance comparison with the baseline methods in Figure 22. In this comparison, the performances of both sets of hyperparameters are competitive, with the unsupervised optimization showing even better performance in some cases, thereby demonstrating the effectiveness of our search strategy and confirming the robustness of hyperparameter sensitivity.

Table 11: Optimized hyperparameter settings for Spotscape

| Type | Dataset | $\lambda_{PCL}$ | $\lambda_{SS}$ | $\tau$ | Learning Rate |
|---|---|---|---|---|---|
| Single | DLPFC Patient 1 | 0.0005 | - | 0.75 | 0.00005 |
| Single | DLPFC Patient 2 | 0.0005 | - | 0.75 | 0.0001 |
| Single | DLPFC Patient 3 | 0.0005 | - | 0.75 | 0.0001 |
| Single | MTG Control | 0.0005 | - | 0.75 | 0.0005 |
| Single | MTG AD | 0.0005 | - | 0.75 | 0.0001 |
| Single | Mouse Embryo | 0.0005 | - | 0.75 | 0.0001 |
| Single | NSCLC | 0.0005 | - | 0.75 | 0.0001 |
| Multi Integration | DLPFC | 0.0005 | 10 | 0.75 | 0.0005 |
| Multi Integration | MTG | 0.0005 | 10 | 0.75 | 0.001 |
| Multi Alignment | Mouse Embryo | 0.0005 | 10 | 0.75 | 0.0005 |
| Multi Alignment | Breast Cancer | 0.0005 | 10 | 0.75 | 0.0005 |

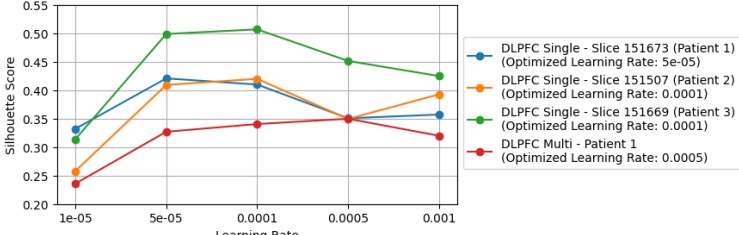

Figure 21: Unsupervised hyperparameter searching strategy using silhouette scores.

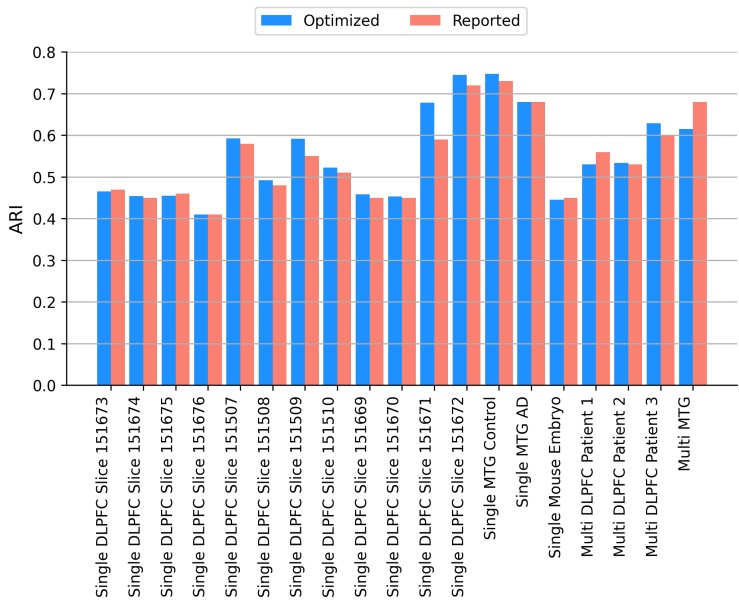

Figure 22: Performance comparison between optimized and reported hyperparameters.

### E.3 IMPLEMENTATION DETAILS

**Model architecture and training.** The model employs a 2-layer GCN (Kipf & Welling, 2016) as the GNN-based encoder and a 2-layer MLP as the decoder, both utilizing batch normalization and ReLU activation functions. The encoder's hidden dimensions are set to $[N_g, 256, 64]$, while the decoder's dimensions are configured as $[64, 256, N_g]$. The clustering process in PCL is performed $T = 3$ times, with the $K$-means granularity set to $[K, 1.5K, 2K]$ to get a fine-grained representation. Optimization is carried out using the Adam optimizer with a learning rate determined through hyperparameter searching (see Appendix E.1) and a weight decay of 0.0001. All experiments are repeated 10 times, and we report the mean and standard deviation for each performance metric.

**Preprocessing.** We follow the preprocessing methodology described in prior work (Dong & Zhang, 2022). Initially, 5000 highly variable genes are selected using Seurat v3 (Stuart et al., 2019). The

data is then normalized to a CPM target of $10,000$ and log-transformed using the SCANPY package (Wolf et al., 2018a). For datasets with multiple slices, we concatenate the slices to enable integration or alignment.

**Computational Resources.** All the experiments are conducted on Intel Xeon Gold 6326 CPU and NVIDIA GeForce A6000 (48GB).

**Software Configuration.** Spotscape is implemented in Python 3 (version 3.9.7) using Py-Torch 2.1.1 (`https://pytorch.org/`) with Pytorch Geometric (`https://github.com/pyg-team/pytorch_geometric`) packages.

## F  SCALABILITY OF SPOTSCAPE

Due to recent advancements in high-throughput sequencing machines, the scalability of models has become a critical factor in validating their performance. To this end, we generate a synthesized dataset by downsampling or oversampling the Mouse Embryo dataset to create data with 1,000 to 100,000 spots, and report the running time in Figure 23. We observed that Spotscape requires relatively more training time than baseline methods due to the prototypical contrastive learning objective. However, the training time of Spotscape scales linearly with the number of spots, rather than quadratically or exponentially. This linear scalability ensures that SpotScape remains practical for high-throughput datasets (e.g., 100,000 spots) within a reasonable timeframe. Moreover, we would like to emphasize that Spotscape without the prototypical learning scheme exhibits faster running times. Thus, if fast inference is required, this option can be used, albeit with a trade-off in performance.

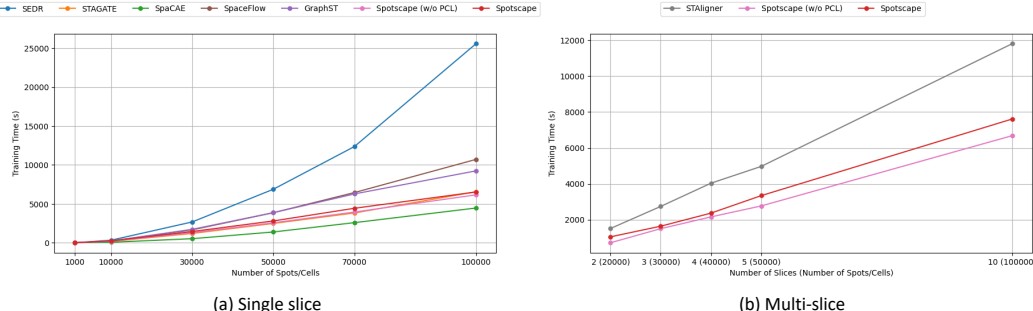

(a) Single slice                     (b) Multi-slice

Figure 23: The running time of Spotscape and baseline methods over the various number of spots on (a) the single and (b) multi-slice dataset.

# G    DIFFERENTIALLY EXPRESSED GENE ANALYSIS

Genes with $\log_2(\text{fold}) > 0.25$ and adjusted p-value from DESeq2, implemented in FindMarkers from Seurat V4 (Hao et al., 2023) $< 0.05$ are determined as DEGs.

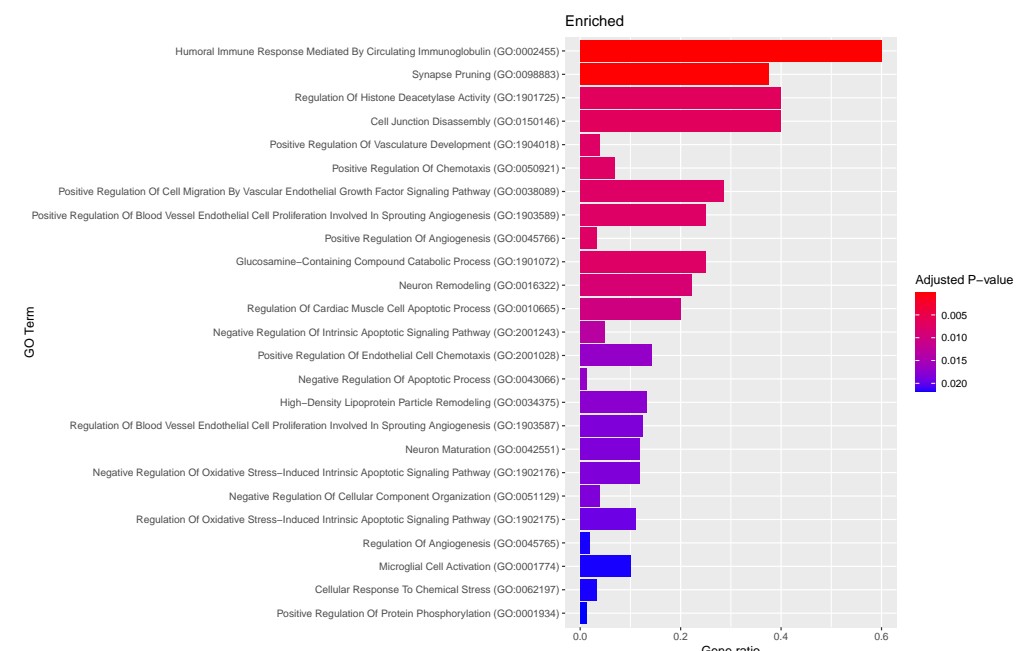

Figure 24: Differential gene analysis and gene ontology enrichment analysis for biological process between AD and PSP in cluster 6 (layer 2).

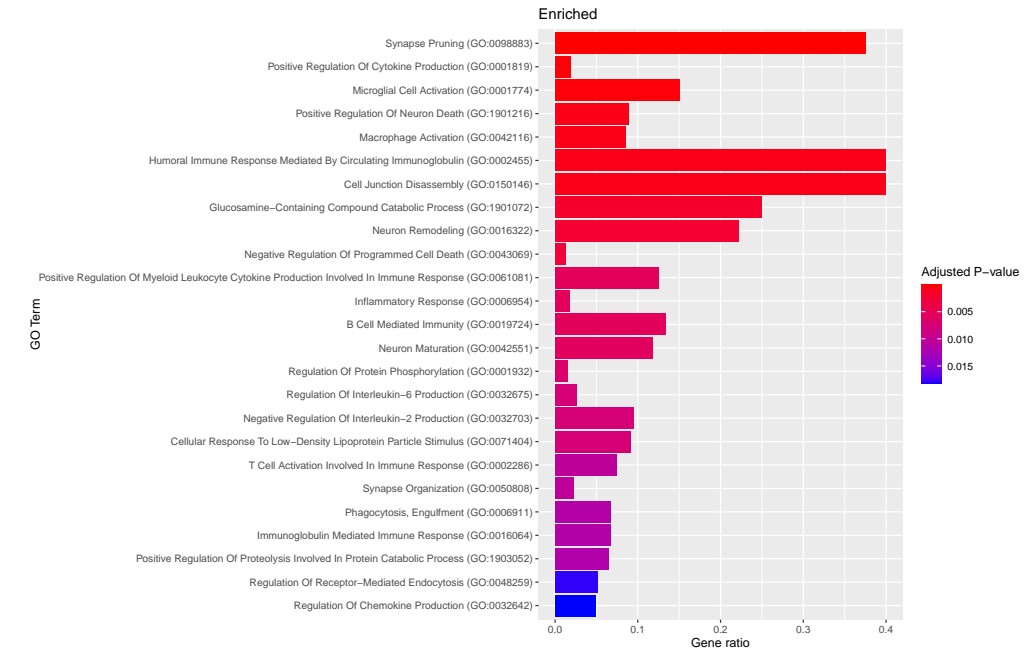

Figure 25: Differential gene analysis and gene ontology enrichment analysis for biological process between AD and PSP in cluster 4 (layer 5).

# H    TRAJECTORY ANALYSIS

We perform trajectory inference tasks to evaluate whether the representation learned by Spotscape effectively captures underlying trajectories in spatial transcriptomic data. For quantitative validation, we assign numerical values to layers as follows: WM = 0, layer 6 = 1, layer 5 = 2, layer 4 = 3, layer 3 = 4, layer 2 = 5, and layer 1 = 6. We then calculate pseudo-Spatiotemporal Map (pSM) values following the approach described in SpaceFlow Ren et al. (2022) using the representation from each model. Finally, we compute the correlation between these assigned values and the calculated pSM values and report the results in Figure 26. In these results, Spotscape demonstrates effectiveness in the trajectory inference task, further validating its broad applicability. Additionally, it is worth noting that while Spotscape employs a prototypical contrastive learning scheme that could make the latent space discrete, potentially negatively affecting the trajectory inference task, Spotscape is not dominated by this module and still demonstrates strong performance as long as the balance coefficient ($L_{Pro}$) is not set too high. We also present these results visually in Figure 27.

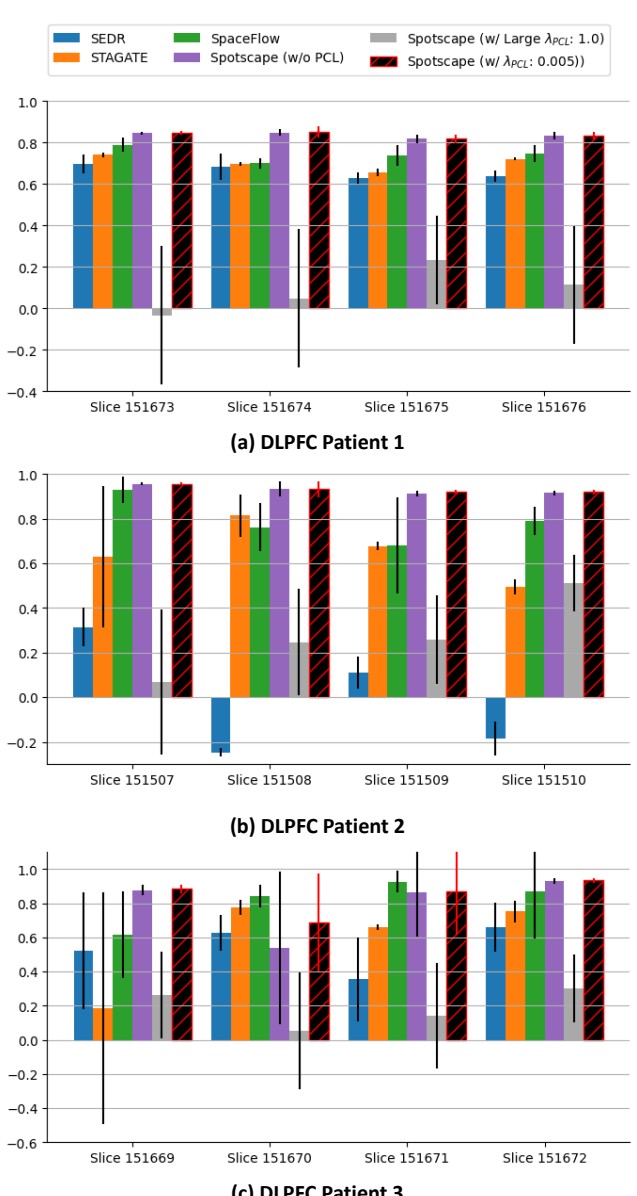

Figure 26: Correlation Coefficient between pseudo-Spatiotemporal Map and Layers in DLPFC.

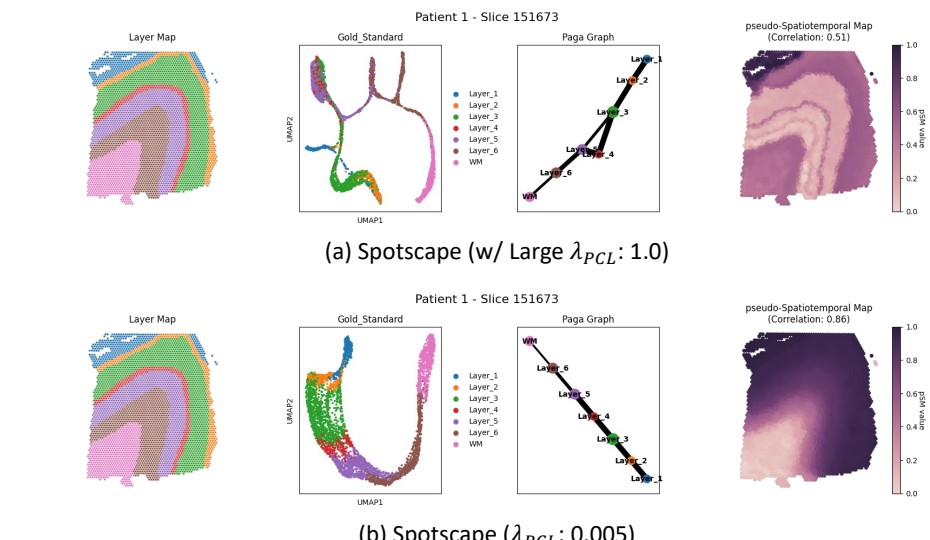

(a) Spotscape (w/ Large $\lambda_{PCL}$: 1.0)

(b) Spotscape ($\lambda_{PCL}$: 0.005)

Figure 27: Trajectory inference results of Spotscape.

# I IMPUTATION

To demonstrate the additional benefits of incorporating a decoder layer and reconstruction loss, we performed imputation tasks to highlight the effectiveness of our reconstructed output in imputing missing values and denoising noise present in the raw data. In the experiment shown in Figure 28, we masked certain non-zero values in the data and evaluated whether the model successfully recovers these values, following the settings from previous works Lee et al. (2024). From these results, Spotscape outperforms in terms of both RMSE and median L1-distance, demonstrating its superiority in imputation tasks. Moreover, we also examine whether the imputed outputs can help identify marker genes that were not differently expressed in the raw data, illustrated in Figure 29. We conduct these experiments for the known marker genes in the brain cortex layer. Specifically, RORB serves as a canonical marker for layer 4 neurons (Clark et al., 2020); ETV1 is associated with layer 5 neurons (Goralski et al., 2024); NTNG2 and NR4A2 are well-recognized markers for layer 6 neurons (Maynard et al., 2021; Darbandi et al., 2018); and OLIG2 is indicative of white matter regions (Wegener et al., 2015). The results show that after imputation using Spotscape, marker genes are more distinctly expressed, demonstrating the practical applicability of Spotscape.

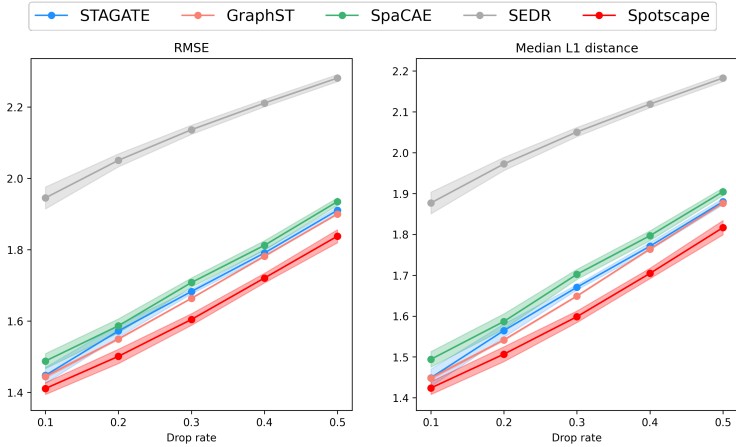

Figure 28: Imputation error comparison across various drop rates in the DLPFC.

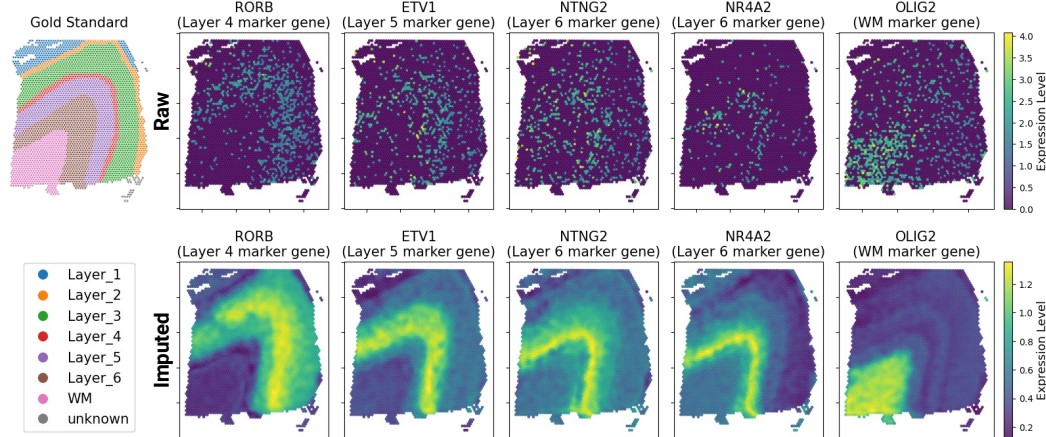

Figure 29: Spatial expression of raw and Spotscape imputed data for marker genes in the DLPFC.

## J  FUTURE WORKS

In this work, we discover that reflecting the global relationships between spots provides significant information on SRT data; however, we currently leverage this relationship only implicitly through the loss function. We recognize that the model could benefit from incorporating more complex interactions by constructing edges between spots, thereby implementing graph structure learning. Future work could explore this avenue to enhance the representation of spatial relationships, allowing the model to leverage valuable information from the global context more effectively.

Furthermore, SRT data frequently includes histology images that offer critical contextual information about tissue architecture and cellular organization. However, in this study, we concentrate on a more general case that limits our analysis to spatial coordinates and gene expression profiles, potentially overlooking the rich insights that histological features could provide. We anticipate that integrating this information with Spotscape could represent a promising direction for future research.

