# OpenReview forum: "Global Context-aware Representation Learning for Spatially Resolved Transcriptomics"
_ICLR.cc/2025/Conference — Submitted to ICLR 2025_

### Official Review · Reviewer_TZi9 · 2024-10-31

**Soundness:** 3
**Presentation:** 3
**Contribution:** 3
**Rating:** 6
**Confidence:** 4

**Summary:**

The paper introduces Spotscape, a framework designed to enhance the analysis of spatially resolved transcriptomics (SRT) data. Traditional graph-based deep learning methods in SRT have limitations in representing spots near the boundaries of cell type clusters, as they tend to focus mainly on spatially close spots with similar features. Spotscape addresses this by introducing a Similarity Telescope module, which captures global relationships among multiple spots to improve representation. Moreover, to handle the integration of multiple slices from diverse sources, the framework incorporates a similarity scaling strategy that maintains consistent distances between intra- and inter-slice spots. Experiments show that Spotscape outperforms existing methods in tasks such as spatial domain identification, multi-slice integration, and alignment.

**Strengths:**

1. It is quite impressive that Spotscape can handle both single-slice and multi-slice tasks, from spatial domain detection to alignment of mouse embryogenesis.
2. The experimental results show that Spotscape delivers performance boost compared to the baselines in different tasks.
3. The ablation study showcases the effects of each type of loss in different tasks, helping the readers to better understand the design of Spotscape.

**Weaknesses:**

1. The provided information is not enough to reproduce the experiments. For example, what kind of message passing layer is used in the GNN encoder? How many GNN layers? What are the dimensions of the layers? What are the choices for hyperparameters?
2. The proposed Spotscape looks more complicated than the baselines. I wonder if is Spotscape more resource-consuming in terms of spatial and computation complexity.
3. (minor) There are some typos in the manuscript. For example, the title of Table 2.

**Questions:**

1. I appreciate the authors for providing the data visualizations based on spatial coordinates. Notice that Spotscape emphasizes a lot on feature similarities, which I suppose is the key to performance improvement. It would be great to have some additional figures on the similarities regarding expression profiles. For example, a UMAP plot for the MTG data shows that the same cell types in control and AD group share similar gene expression.
2. It is great that the authors showcase some interesting alignment results with mouse embryo slices and cross-technology alignment. For the mouse embryo slices, a more general baseline is (unbalanced) optimal transport, which typically performs pretty good in temporal alignment. The results for Xenium and Visium alignment are not quite persuasive, since only a very small part of the slice is used for evaluation.

---

> ### Author Response · Authors · 2024-11-23
> **(1/N) Author Rebuttal**
>
> Thank you so much for your thorough and insightful feedback!
>
> > The provided information is not enough to reproduce the experiments. For example, what kind of message passing layer is used in the GNN encoder? How many GNN layers? What are the dimensions of the layers? What are the choices for hyperparameters?
>
> We apologize for the lack of detail regarding the hyperparameters and implementation in the original manuscript, and we appreciate your feedback.
>
> Spotscape employs a 2-layer GCN as the GNN-based encoder and a 2-layer MLP as the decoder, both utilizing batch normalization and ReLU activation functions. The encoder’s hidden dimensions are configured as [256, 64], while the decoder’s dimensions are set to [64, 256]. These implementation details can be found in Appendix E.3.
>
> Additionally, regarding hyperparameter selection, while some studies in this field report baseline method performance using default hyperparameters to reflect the unsupervised nature of tasks such as clustering or alignment, variations in datasets and preprocessing steps across studies may lead to an underestimation of baseline performance under default settings. To ensure a fair and reliable comparison, we conducted an extensive hyperparameter search for both Spotscape and baseline methods. This process systematically optimized test performance by exploring all loss-balancing parameters and learning rates. Details of this search, including hyperparameters for baseline models, are provided in Appendix E.1.
>
> Furthermore, to address concerns about potential overfitting during hyperparameter tuning, we performed a detailed sensitivity analysis in Appendix D. The analysis revealed that, apart from the learning rate, the hyperparameters of Spotscape are robust across reasonable ranges, with minimal impact on performance. Building on this, we introduced an unsupervised hyperparameter optimization strategy in Appendix E.2. Fixing all hyperparameters except the learning rate, we conducted a search over values {0.001,0.0005,0.0001,0.00005,0.00001} based on the silhouette score, computed in an unsupervised manner. The results are summarized in Table 11, with corresponding silhouette scores for the DLPFC dataset presented in Figure 21.
>
> Finally, we compared the performance achieved using hyperparameters selected through unsupervised optimization with those obtained via supervised NMI-based selection. As shown in Figure 22, both approaches produced comparable results, highlighting the robustness of our hyperparameter tuning strategy and the generalizability of Spotscape across diverse evaluation methods. We hope these clarifications address your concerns and provide the necessary information for reproducibility.

---

> ### Author Response · Authors · 2024-11-23
> **(2/N) Author Rebuttal**
>
> > The proposed Spotscape looks more complicated than the baselines. I wonder if is Spotscape more resource-consuming in terms of spatial and computation complexity.
>
> Thank you for your insightful feedback, which provides an excellent opportunity to enhance our paper.
>
> To address this concern, we compare the training speed of Spotscape with baseline methods in Appendix F, using synthetic datasets ranging from 1,000 to 100,000 spots. As expected, Spotscape requires slightly longer training times than the baselines, primarily due to the prototypical contrastive learning (PCL) objective. However, we observed that Spotscape’s training time scales linearly with the number of spots, rather than quadratically or exponentially. This linear scalability ensures that Spotscape remains feasible for large-scale datasets, such as those with 100,000 spots, while still completing within a reasonable timeframe.
>
> While faster computation is always desirable, it is important to note that this domain emphasizes achieving higher accuracy over real-time inference, unlike domains such as NLP, where online responsiveness is often critical. Moreover, we provide an alternative configuration of Spotscape without the PCL objective, which significantly reduces computational complexity and outperforms most baselines. However, as shown in Figure 7, this simplified version incurs a trade-off in performance. This makes it a practical option for scenarios where faster inference is prioritized over optimal accuracy.
>
> We hope this detailed analysis clarifies that while Spotscape is moderately more resource-intensive, its linear scalability and adaptability make it suitable for a wide range of datasets and use cases.

---

> ### Author Response · Authors · 2024-11-23
> **(3/N) Author Rebuttal**
>
> > There are some typos in the manuscript. For example, the title of Table 2.
>
> Thank you for your careful review and for pointing out the typos. We have addressed this by splitting Table 2 into two separate tables, organized by datasets: MTG and Mouse Embryo. Additionally, we revised the caption of Table 2 to enhance clarity and accuracy. Your feedback has been invaluable in improving the quality of the manuscript.

---

> ### Author Response · Authors · 2024-11-23
> **(4/N) Author Rebuttal**
>
> > I appreciate the authors for providing the data visualizations based on spatial coordinates. Notice that Spotscape emphasizes a lot on feature similarities, which I suppose is the key to performance improvement. It would be great to have some additional figures on the similarities regarding expression profiles. For example, a UMAP plot for the MTG data shows that the same cell types in control and AD group share similar gene expression.
>
> Thank you for acknowledging our efforts in providing data visualizations based on spatial coordinates. We greatly appreciate your insightful suggestion to further explore learned feature similarities.
>
> To provide an additional perspective on learned feature similarity, we conducted pseudotime analysis, as detailed in Appendix H.
> Pseudotime is a computational metric used to infer the relative progression of cells along a biological trajectory. Instead of measuring actual time, pseudotime orders cells based on their gene expression profiles, reflecting their position in a biological process (e.g., differentiation).
>  By combining spatial data with trajectory analysis, we can uncover spatial gradients of gene expression.
>  In the experiments presented in Figure 26, we quantitatively assess the pseudotime performance by computing the correlation between pseudo-Spatiotemporal Map (pSM) values and gold standard spatial regions.
> Our results demonstrate that Spotscape outperforms baseline methods, indicating that the learned feature similarities effectively capture the continuous nature of gene expression across spatial regions.
>
> Additionally, to address your specific suggestion, we included a UMAP plot and visualize trajectory inference results for the single DLPFC slice, shown in Figure 27. This visualization highlights how gene expression similarities correspond to spatial domains, further illustrating the effectiveness of Spotscape in capturing feature similarities.

---

> ### Author Response · Authors · 2024-11-23
> **(N/N) Author Rebuttal**
>
> > It is great that the authors showcase some interesting alignment results with mouse embryo slices and cross-technology alignment. For the mouse embryo slices, a more general baseline is (unbalanced) optimal transport, which typically performs pretty good in temporal alignment.
>
> Thank you for the valuable suggestion. To address this, we conducted experiments using (unbalanced) optimal transport, implemented with the Python Optimal Transport (POT) library.
>
> |               | **LTARI** |
> |:--------------|:---------:|
> | Unbalanced OT | 0.06      |
> | STAligner     | 0.46      |
> | scSLAT        | 0.52      |
> | **Spotscape** | **0.56**  |
>
> In our experiments, O.T demonstrated extremely low alignment performance. We attribute this to its inability to adequately incorporate spatial information, which is critical for achieving accurate alignment in this domain.
>
> > The results for Xenium and Visium alignment are not quite persuasive, since only a very small part of the slice is used for evaluation.
>
> While it is true that only a small number of spots were used in the evaluation, we emphasize that this reflects the challenging nature of the task, as alignment must accurately match specific subsets of spots among many. This highlights the complexity of aligning sparse regions in spatial transcriptomics data.
>
> Furthermore, we have already demonstrated the quantitative performance of our approach in Table 7, which validates the effectiveness of our method. Our intention in showcasing this experiment was to present a practical and realistic case of aligning cancer cell types, a critical application scenario in this field.

---

> > ### Comment · Reviewer_TZi9 · 2024-11-25
> > **Thank you for the rebuttal**
> >
> > Thank you for the detailed rebuttal and responses. The proposed method introduces some computational overhead, but the significant performance improvements achieved are commendable and desirable. I have a few follow-up questions and comments based on your replies:
> >
> > 1. While I understand the time constraints of the rebuttal period, I strongly encourage the addition of error bars to Fig. 26 in a future revision. Without them, the figure lacks statistically meaningful information and limits interpretability.
> >
> > 2. In Fig. 26, the parameter $\lambda_{PCL}$ appears to have a substantial impact on the performance of Spotscape, with large values of $\lambda_{PCL}$ leading to significant performance degradation. This trend aligns with the hyperparameter choices outlined in Appendix E. Could you provide a more detailed explanation of why large $\lambda_{PCL}$ values adversely affect performance?
> >
> > 3. In addition to bullet 2, the sensitivity analysis in Appendix D reveals that Spotscape is also sensitive to the choices of $\lambda_{Recon}$ and $\lambda_{SC}$. This observation suggests that the performance of Spotscape is heavily influenced by the balance and interplay between the different loss components. Could you provide additional discussion or insights on how these components interact and why certain combinations may be particularly effective?

---

> > > ### Author Response · Authors · 2024-11-28
> > > **Official Comment by Authors**
> > >
> > > > While I understand the time constraints of the rebuttal period, I strongly encourage the addition of error bars to Fig. 26 in a future revision. Without them, the figure lacks statistically meaningful information and limits interpretability.
> > >
> > > Thank you for your understanding and feedback. We acknowledge the importance of including error bars in Figure 26 to enhance its statistical validity. We ran it 10 times and reported the average performance along with its standard error. The results demonstrate that SpotScape consistently outperforms other baseline methods with minimal standard error, highlighting its robustness. While we initially anticipated that the prototypical loss might adversely affect trajectory inference by disrupting the continuity of the latent space, this concern was mitigated due to the loss being assigned a relatively small weight ($\lambda_{PCL}$ = 0.005), preventing it from dominating the representation learning process. However, such adverse effects can occur when the weight ($\lambda_{PCL} = 1$) is set too high.
> > >
> > > > In Fig. 26, the parameter $\lambda_{PCL}$ appears to have a substantial impact on the performance of Spotscape, with large values of $\lambda_{PCL}$ leading to significant performance degradation. This trend aligns with the hyperparameter choices outlined in Appendix E. Could you provide a more detailed explanation of why large $\lambda_{PCL}$ values adversely affect performance?
> > >
> > > Thank you for your insightful question regarding the impact of large $\lambda_{PCL}$ values on Spotscape's performance.
> > >
> > > The prototypical contrastive learning (PCL) term, controlled by $\lambda_{PCL}$, is designed to bring semantically similar spots closer in the latent space, based on the assumption that spots within the same cluster share similar semantics. This mechanism enhances representation learning by pulling semantically similar spots closer and pushing dissimilar ones apart.
> > >
> > > In this context, $\lambda_{SC}$ plays a crucial role in learning robust spot representations in a self-supervised manner by consistently preserving underlying similarities, regardless of graph augmentation. The PCL term complements this process by refining the learned representations, similar to self-training. However, when $\lambda_{PCL}$ is assigned a large weight, it can dominate other loss terms, potentially leading to sampling bias by pulling spot representations that are not semantically similar. Ultimately, this leads to a degradation in performance as the learned representations fail to generalize effectively.
> > >
> > > >In addition to bullet 2, the sensitivity analysis in Appendix D reveals that Spotscape is also sensitive to the choices of $\lambda_{Recon}$ and $\lambda_{SC}$. This observation suggests that the performance of Spotscape is heavily influenced by the balance and interplay between the different loss components. Could you provide additional discussion or insights on how these components interact and why certain combinations may be particularly effective?
> > >
> > > As noted in the sensitivity analysis section (Appendix D), Spotscape exhibits performance degradation when $\lambda_{SC}$ is low or $\lambda_{Recon}$ is high. These results align with the intuition behind the design of Spotscape. The primary contribution of Spotscape lies in learning global context similarities while maintaining consistency across different augmentations. To achieve this, a relatively high weight for $\lambda_{SC}$ is necessary, as it ensures effective representation learning. Conversely, $\lambda_{Recon}$ serves an auxiliary role to prevent degenerate solutions, requiring a comparatively lower weight. **Based on these considerations, we set $\lambda_{SC} = 1$ and $\lambda_{Recon} = 0.1$, assigning $\lambda_{SC}$ a weight ten times greater than that of $\lambda_{Recon}$.** This balance enables Spotscape to emphasize its core learning objectives while avoiding overfitting to reconstruction constraints.
> > >
> > > Additionally, as addressed in the previous question, the prototypical contrastive loss ($\lambda_{PCL}$) plays a supporting role in enhancing spot representations by bringing semantically similar spots closer in the latent space. However, assigning too high a weight to $\lambda_{PCL}$ can introduce sampling bias and negatively impact performance. **To mitigate this, we set $\lambda_{PCL} = 0.0005$, which is significantly smaller than the main self-supervised representation learning components, ensuring it contributes without dominating the overall loss.**
> > >
> > > I hope this follow-up discussion has addressed any remaining ambiguities.

---

> ### Author Response · Authors · 2024-11-30
> **Gentle Reminder**
>
> **Dear Reviewer TZi9,**
>
> Thank you for your thoughtful and constructive review. We sincerely appreciate the time and effort you’ve invested in providing such valuable feedback. We are pleased that our previous responses have addressed many of your concerns.
>
> With regard to your new comments on the writing aspects, we would be grateful if you could confirm whether our revised explanations sufficiently address your points regarding
> * the statistical significance of the information in Fig. 26, as well as
> * the balance parameters ($\lambda_{PCL}$, $\lambda_{Recon}$, and $\lambda_{SC}$).
>
> If any questions or concerns remain, we would be happy to continue the discussion and provide further clarifications.
>
> Once again, thank you for your valuable contributions to our manuscript. Your insights have played a crucial role in enhancing the overall quality and clarity of our work. We look forward to your feedback and reevaluation of our work.
>
> Best regards,
> The Authors

---

> ### Comment · Reviewer_TZi9 · 2024-12-02
> **Thank you for the response**
>
> Thank you for providing the additional results and discussions. The proposed method involves multiple objectives and requires carefully handcrafted designs to balance them, which potentially limits the generalizability of it. However, I believe the main advantage of this work lies in its broad range of application scenarios and its impressive performance. These advantages outweigh the limitations of model design. Therefore, I will maintain my score to support this manuscript.

---

> ### Author Response · Authors · 2024-12-02
> **Official Comment by Authors**
>
> Thank you very much for your detailed review and for highlighting both the strengths and limitations of our work. We greatly appreciate your acknowledgment of the broad applicability and strong performance of our method.
>
> Regarding your concern about the handcrafted design for balancing multiple objectives, we would like to point out an important result from Figure 22. **Specifically, Spotscape demonstrates that even when using a fixed balance parameter (without additional search), the performance remains comparable to the results achieved with parameter search across all 19 datasets.** This indicates that our method can maintain robust performance without requiring extensive tuning, which mitigates concerns about its generalizability.
>
> We hope this clarification further emphasizes the practicality and adaptability of Spotscape in diverse scenarios. Thank you again for your constructive feedback and support for our manuscript.

---

### Official Review · Reviewer_uUBu · 2024-11-01

**Soundness:** 2
**Presentation:** 2
**Contribution:** 2
**Rating:** 6
**Confidence:** 3

**Summary:**

This paper studies the representation learning for spatial transcriptomics data. The authors argue that capturing global context is critical, especially for spots near boundaries. To address it, they propose pretraining the GNNs using contrastive learning to align the representations between inter- and intra-slice spots. Several experiments including spatial domain identification, multi-slice integration, and alignment, demonstrate the model’s effectiveness.

**Strengths:**

- The preliminary findings showing low clustering performance for boundary spots are interesting and can potentially motivate some future studies to address such an issue.
- The experiment is comprehensive, including multiple datasets across different sequencing technologies and patient samples. Additionally, aligning slides sequenced by Visium and Xenium is both practically significant and challenging due to batch effects.
- For methodology, the paper demonstrates the effectiveness of graph contrastive learning in spatial transcriptomics data.

**Weaknesses:**

- The motivation is to capture global context through graph contrastive learning, but they haven't justified it. It’s not surprising that graph contrastive learning can improve GNN’s performance [1,2,3]. The clustering improvements for boundary spots in Figure 1 likely result from enhanced representation across all spots, as non-boundary spots also show significant improvement. Besides, aligning representations across graph augmentations doesn’t inherently enable access to information beyond local neighborhoods.
- In the methodology, several alignment loss functions are proposed, but there appears to be redundancy. For example, both L_{sc} (Equ.(1)) and L_{pcl} aim to maximize the representation similarity between two augmentations. Additionally, as shown in Figure (7), removing L_{recon} and L_{pcl} doesn’t actually impact the performance, raising the question of whether so many loss functions are necessary.
- Several experimental details are missing, including the type of graph augmentation used, the specific hyperparameters, and the tuning strategy applied.
- The explanation of Section 4.4 can be improved. The “A” is first introduced in Section 3, but it is reintroduced with a different explanation in Line 287, leading to potential confusion.

[1] Graph Contrastive Learning with Augmentations, NeurIPS2020

[2] BatchSampler: Sampling Mini-Batches for Contrastive Learning in Vision, Language, and Graphs, KDD2023

[3] Deep Graph Contrastive Representation Learning

**Questions:**

Downstream tasks are mainly about the clustering alignment. Could the author evaluate the model on some other clinical downstream tasks?

---

> ### Author Response · Authors · 2024-11-23
> **(1/N) Author Rebuttal**
>
> Thank you for giving us valuable feedback!
>
> > The motivation is to capture global context through graph contrastive learning, but they haven't justified it. It’s not surprising that graph contrastive learning can improve GNN’s performance [1,2,3]. The clustering improvements for boundary spots in Figure 1 likely result from enhanced representation across all spots, as non-boundary spots also show significant improvement. Besides, aligning representations across graph augmentations doesn’t inherently enable access to information beyond local neighborhoods.
>
> To address your concerns regarding our paper, we will respond to your points sequentially, starting from the last sentence and moving in reverse order.
>
> > Besides, aligning representations across graph augmentations doesn’t inherently enable access to information beyond local neighborhoods.
>
> As you mentioned, simply aligning representations between two differently augmented graphs cannot inherently access information beyond local neighborhoods. This limitation arises because the receptive field of a vanilla GCN is restricted to the local neighborhood, and alignment loss functions like $MSE(\tilde{Z}, \tilde{Z}^{'})$ only focus on the difference $\tilde{Z_{i}} - \tilde{Z_{i}^{'}}​$ for each individual node i.
>
> However, our proposed **relation consistency loss** in equation 1 goes beyond local neighborhoods by capturing global relationships. Specifically, it considers relationships between pairs of nodes, even those outside the local receptive field. This is achieved by comparing terms like $\tilde{Z}_i \cdot \tilde{Z}^{'}_j$ and $\tilde{Z}^{'}_i \cdot  \tilde{Z}_j$, which effectively align similarities between nodes i and j, irrespective of their distance in coordinate space (i.e., graph structure). By incorporating these global relationships, our loss function enables the model to capture global context and optimize node representations accordingly.
>
> The effectiveness of this approach in capturing global similarities is demonstrated in the experiments presented in Figure 8, where Spotscape uniquely reveals continuous similarity gradients and varying levels of similarity across spatial domains. In contrast, without the $L_{SC}$, the model is unable to learn these global relationships, highlighting the essential role of our loss function in leveraging global context for more effective node representation learning.
>
> > The clustering improvements for boundary spots in Figure 1 likely result from enhanced representation across all spots, as non-boundary spots also show significant improvement.
>
> We agree with your observation that enhanced representations benefit both boundary and non-boundary spots. Spotscape’s ability to capture global context contributes to improving representations across all spots, resulting in superior clustering performance for both types.
>
> > It’s not surprising that graph contrastive learning can improve GNN’s performance [1,2,3].
>
> While graph contrastive learning is known to enhance GNNs, our method differs significantly from existing approaches as stated above. To validate our approach, we compare Spotscape with existing self-supervised learning methods in Figure 13 of Appendix B. The results demonstrate that Spotscape achieves significantly better performance, further verifying the unique contributions of our method.

---

> ### Author Response · Authors · 2024-11-23
> **(2/N) Author Rebuttal**
>
> > In the methodology, several alignment loss functions are proposed, but there appears to be redundancy. For example, both $L_{SC}$ (Equ.(1)) and $L_{PCL}$ aim to maximize the representation similarity between two augmentations. Additionally, as shown in Figure (7), removing $L_{Recon}$ and $L_{PCL}$ doesn’t actually impact the performance, raising the question of whether so many loss functions are necessary.
>
> As noted in the ablation study, the reconstruction loss ($L_{Recon}$) does not have a significant impact on performance in our datasets. However, it plays a preventive role in addressing a well-known issue in Siamese networks: the risk of degenerate solutions, where all representations collapse into a single point. While this issue did not arise in our specific datasets, including $L_{Recon}$ ensures robustness and prevents potential degeneration in other datasets or scenarios.
>
> An additional benefit of using the reconstruction layer is that it provides a denoised matrix, which can be applied to downstream tasks by imputing missing values and reducing noise in the raw data. To demonstrate this, in Figure 28 of Appendix I, we show that Spotscape outperforms baseline methods in imputing missing values. Additionally, in Figure 29, we demonstrate that known marker genes in brain cortex layers, which are not highly expressed in the raw data, become distinguishable in the reconstructed output. This highlights the potential of the reconstruction component to enable additional tasks.
>
> In the case of the prototypical loss ($L_{PCL}$), while the relation consistency loss ($L_{SC}$​) effectively aligns representations from two differently augmented graphs to learn spot representations, the prototypical contrastive learning​ scheme further enhances these representations by clustering semantically similar spots around their respective prototypes.
>
> As shown in the ablation study figure, $L_{PCL}$​ provides modest performance improvement in the single-slice SDI task, which relatively requires less representation enhancement. However, in more complex tasks such as multi-slice integration and alignment, L_PCL​ shows substantial performance gains, with ARI improving from 0.6068 to 0.6801 (approximately 12.1% increment) and LTARI increasing from 0.5495 to 0.5605 (approximately 2.0% increment). These results highlight the pivotal role of $L_{PCL}$​ in enhancing representation quality for challenging scenarios.
>
> To further demonstrate this, we conducted a new experiment integrating three slices from each of four patients, totaling twelve slices in the DLPFC data. In this much more difficult task, the absence of the prototypical loss resulted in a much larger performance gap, verifying the importance of $L_{PCL}$​ .
>
> | w/o $L_{Recon}$ | w/o $L_{SC}$ | w/o $L_{PCL}$ | w/o $L_{SS}$ | Spotscape |
> |----------|----------|----------|----------|----------|
> | 0.2526 (0.2289) | 0.3350 (0.0885) | 0.1847 (0.0600) | 0.3823 (0.0238) | **0.4461 (0.0842)** |

---

> ### Author Response · Authors · 2024-11-23
> **(3/N) Author Rebuttal**
>
> > Several experimental details are missing, including the type of graph augmentation used, the specific hyperparameters, and the tuning strategy applied.
>
> * Response regarding graph augmentations
>
> Thank you for pointing out the need for additional clarification. In Section 4.1, we noted that the graph augmentations used in our experiments include node feature masking and edge masking, which are standard techniques in graph self-supervised learning. However, we acknowledge that the specific masking ratios were not provided. We have now included this information in Table 10 for greater transparency.
>
> * Response regarding hyperparameter selection and tuning strategy
>
> We apologize for not clarifying the hyperparameter selection in the previous manuscript.
>
> While some studies in this domain report the performance of baseline methods using default hyperparameters, reflecting the unsupervised nature of tasks like clustering or alignment, the datasets and preprocessing steps often vary across studies.
> This can lead to underestimation of baseline performance when default parameters are applied. To ensure a fair comparison, we performed a comprehensive hyperparameter search for both our model and the baseline methods, optimizing for test performance.
> This approach ensures a more accurate and fair evaluation of all methods.
> Additionally, we searched all balance parameters and learning rates for each baseline model, including any additional parameters needed for specific baselines. A detailed description of the search space and the final selected parameters is provided in Appendix Section E.1.
>
> Furthermore, to address concerns regarding potential overfitting to the test data through hyperparameter tuning, we conducted a thorough sensitivity analysis.
> As shown in Appendix Section D, we found that, except for the learning rate, Spotscape’s hyperparameters are robust across reasonable values, meaning their selection has minimal impact on performance.
> Building on this, in Appendix Section E.2, we fixed all hyperparameters except the learning rate and optimized it from the set {0.001, 0.0005, 0.0001, 0.00005, 0.00001} based on the silhouette score, which is computed in an unsupervised manner.
> The results of this search are shown in Table 11, with corresponding silhouette scores for the DLPFC data in Figure 21.
>
> Moreover, in Figure 22, we compared the performance achieved using these unsupervised-selected hyperparameters with that from supervised NMI-based selection and found that both approaches yield similar results. This demonstrates the effectiveness of our search strategy and confirms the robustness of hyperparameter sensitivity.

---

> ### Author Response · Authors · 2024-11-23
> **(4/N) Author Rebuttal**
>
> > The explanation of Section 4.4 can be improved. The “A” is first introduced in Section 3, but it is reintroduced with a different explanation in Line 287, leading to potential confusion.
>
> Thank you for pointing out this potential source of confusion. To address this, we have revised the notation in Equation 6. Specifically, we have replaced $A$, and $B$ with $S_{top}^{(c)}$ and $S_{top}^{(j)}$, respectively, to ensure consistency and clarity throughout the paper.

---

> ### Author Response · Authors · 2024-11-23
> **(N/N) Author Rebuttal**
>
> > Downstream tasks are mainly about the clustering alignment. Could the author evaluate the model on some other clinical downstream tasks?
>
> We appreciate the reviewer’s comment and agree that evaluating our model on additional clinical downstream tasks could be valuable.
>
> In the context of spatial transcriptomics, one of the most direct clinical downstream tasks is investigating the biological relevance of clusters through the identification of enriched pathways or biological processes. To this end, we performed Gene Ontology (GO) enrichment analysis on differentially expressed genes (DEGs) between the control and Alzheimer's disease (AD) groups, and the results are reported in lines 461–479. We found highly enriched biological processes in Layer 2 and Layer 5, as shown in Figures 24 and 25, respectively, most of which are strongly associated with AD. These findings directly address the clinical relevance of our results.
>
> Furthermore, please kindly note that our paper addresses a broad range of tasks, including clustering, alignment, integration, imputation, enrichment analysis, and trajectory inference. This is a significant strength of our work, especially when compared to most previous studies that focus on only one or two tasks.

---

> ### Author Response · Authors · 2024-11-30
> **Gentle Reminder**
>
> **Dear Reviewer uUbu,**
>
> Thank you for your insightful review and for dedicating your time to provide valuable feedback. In response to your suggestions, we have clarified our statements and included further experiments.
>
> * Additional Experiments:
> 	* Comprehensive hyperparameter selection.
> 	* Additional downstream tasks with gene ontology enrichment analysis, imputation, and trajectory analysis.
> * Clarifications and Updates:
> 	* Global context and clustering improvements.
> 	* Redundancy in Loss Functions.
>
>
> We would greatly appreciate it if you could share any further questions or comments. We look forward to your reevaluation of our work.
>
> Best regards,
> The Authors

---

> > ### Comment · Reviewer_uUBu · 2024-12-02
> >
> > Thank you for your response and the effort put into refining the manuscript to include many important details.
> >
> > While I still feel that the methodology is somewhat incremental, I appreciate the applications presented in the paper. I would like to increase my score and will discuss this further with the AC in the next stage.

---

> > > ### Author Response · Authors · 2024-12-02
> > > **Official Comment by Authors**
> > >
> > > Thank you for acknowledging our efforts! Please let us know if there are any remaining points you'd like to discuss further with us.

---

### Official Review · Reviewer_K25U · 2024-11-06

**Soundness:** 2
**Presentation:** 3
**Contribution:** 3
**Rating:** 5
**Confidence:** 4

**Summary:**

The authors propose a novel framework, Spotscape for spatial domain identification and SRT data integration, which achieves impressive performance. The Spotscape framework introduces a slide-level contrastive learning module to learn the spot representation, a prototypical contrastive learning loss for enhancing clustering, a reconstruction loss to stabilize the optimization procedure, and a similarity scaling strategy that maintains consistency across different tissues.

The paper attempts to address some issues unsolved by previous studies, for example, the suboptimal result for boundary spots, the batch effect across multiple slides. The motivation is clear and sensible.

**Strengths:**

1. The paper studies some prominent issues, such as boundary spots, and batch effect.

2. The paper provides comprehensive experiments. The baseline models, experiment setups, and ablation studies are all carefully selected. The performance of the method is hightly competitive.

3. The paper provides source codes for reproducibility.

**Weaknesses:**

1. The logic of the methodology design is not very clear, some designs are very ad-hoc, which needs further clarification.

2. It is not clear how the hyperparameters lambdas are selected. In the codes provided by the authors, those hyperparameters are manually specified. It is possible that these lambdas are chosen to achieve the best test performance, which is not fair for other baseline methods. The authors need to provide additional hyperparameter-sensitive analysis to justify this.

3. It is not clear how the hyperparameters of the baseline methods are selected. The authors need to show that they made the best efforts to acquire the best performance from the baselines to ensure a fair comparison.

4. The time complexity of this method is not clear. The prototypical contrastive learning objective involves T times of k-means in each training iteration (after 500 epochs), which may dramatically impact the running speed. The author needs to provide justification for the time-performance tradeoff.

**Questions:**

1. The author mentions multiple times the statement that "In biological systems, cells that are spatially close often exhibit strong feature similarity, regardless of their functional characteristics." Is there any literature to justify this? From my knowledge, this statement is misleading. The similarity only appears at the spot level, while at the cell level, for example, the Xenium data, the cell types locally close to each other are actually quite heterogeneous. Tumor micro-environment is one of the few exceptions where the local cells tend to share similar characteristics. I hope the authors are rigid when making statements about biology.

Meanwhile, statements are causing confusion like this "However, these approaches fall short in obtaining qualified spot representations, particularly for those located around the boundary of **cell type clusters**, as they heavily emphasize spatially local **spots** that have minimal feature differences from an anchor node." It mentions "cell type clusters" and "local spots" in the same time, while spots typically contain heterogeneous cell types, and they correspond to **spatial domain** instead of **cell type clusters**.

2. "Prototypical Contrastive Learning" aims to enhance clustering and rare cell-type classification (I'm not sure how this benefits spot-level tasks). The idea is essentially similar to many previously established prototypical losses [1,2,etc]. However, I did not see any references in the paper. Should authors cite those methods or justify why they need to propose a different method?

3. The authors proudly state that "instead of relying on any predictor or stop gradient techniques, Spotscape simplifies the training procedure by employing a reconstruction loss". To me, it is not clear how the reconstruction loss provides a better solution than the other options. It seems like the authors just blindly chose a shortcut for the implementation.

[1] Prototypical Contrastive Learning of Unsupervised Representations, ICLR 2021
[2] Population-level integration of single-cell datasets enables multi-scale analysis across samples, Nature Methods

---

> ### Author Response · Authors · 2024-11-23
> **(1/N) Author Rebuttal**
>
> Thank you for taking the time to review our paper and provide a valuable review.
>
> > The logic of the methodology design is not very clear, some designs are very ad-hoc, which needs further clarification.
>
> We sincerely apologize for any lack of clarity in explaining our methodologies. To address this, we have added a detailed pseudo-code to illustrate the process of our method more clearly in Appendix Section C. If any aspects remain unclear, we kindly ask for your feedback, and we will make every effort to provide further clarification.

---

> ### Author Response · Authors · 2024-11-23
> **(2/N) Author Rebuttal**
>
> > It is not clear how the hyperparameters lambdas are selected. In the codes provided by the authors, those hyperparameters are manually specified. It is possible that these lambdas are chosen to achieve the best test performance, which is not fair for other baseline methods. The authors need to provide additional hyperparameter-sensitive analysis to justify this.
>
> > It is not clear how the hyperparameters of the baseline methods are selected. The authors need to show that they made the best efforts to acquire the best performance from the baselines to ensure a fair comparison.
>
> We apologize for not clarifying the hyper-parameter selection process, thereby inducing confusion about our reported experiment results.
>
> We acknowledge that some studies in this domain report the performance of baseline methods using their default hyperparameters, reflecting the unsupervised nature of tasks such as clustering or alignment.
> However, the datasets used in different studies often vary, and even when the same dataset is used, preprocessing steps can differ, potentially leading to underestimation of baseline performance when default parameters are applied.
> To ensure a fair comparison, we performed a comprehensive hyperparameter search for both our model and the baseline models, optimizing for test performance.
> This approach allows us to provide a more accurate and fair evaluation of all methods.
> Additionally, we search all balance parameters and learning rates for each baseline method, including any additional parameters required for certain baselines. A more detailed description of the search space and finally selected parameters are provided in Appendix Section E.1.
>
> Furthermore, to address potential concerns that our model's performance depends on hyperparameters tuned using test data, we conducted an extensive sensitivity analysis and also suggested a reasonable unsupervised hyperparameter selection strategy.
> From the sensitivity analysis in Appendix Section D, we observed that, aside from the learning rate, all hyperparameters of Spotscape are robust in reasonable space, indicating that their selection has minimal impact on performance.
> Building on this insight, in Appendix Section E.2, we fixed all hyperparameters except the learning rate and selected its optimal learning rate from the set {0.001, 0.0005, 0.0001, 0.00005, 0.00001} based on the silhouette score, which can be obtained unsupervised manner.
> The results of this search are reported in Table 11, with the corresponding silhouette scores for the DLPFC data shown in Figure 21.
> Furthermore, in Figure 22, we compared the performance achieved using these unsupervised-selected hyperparameters with that obtained from supervised NMI-based selection and find that both approaches yield similar results.
> This demonstrates the effectiveness of our search strategy and confirms the robustness of hyperparameter sensitivity.

---

> ### Author Response · Authors · 2024-11-23
> **(3/N) Author Rebuttal**
>
> > The time complexity of this method is not clear. The prototypical contrastive learning objective involves T times of k-means in each training iteration (after 500 epochs), which may dramatically impact the running speed. The author needs to provide justification for the time-performance tradeoff.
>
> Thank you for your valuable feedback, which helps strengthen our paper.
>
> To address this concern, we report the training speed of Spotscape compared to baseline methods in Appendix Section F, across synthetic datasets ranging from 1,000 to 100,000 spots. As anticipated, Spotscape requires relatively more training time than baseline methods due to the prototypical contrastive learning objective. However, we observed that the training time of Spotscape scales linearly with the number of spots, rather than quadratically or exponentially. This linear scalability ensures that Spotscape remains practical for high-throughput datasets (e.g., 100,000 spots) within a reasonable timeframe.
>
> While shorter running times are beneficial, the nature of this domain prioritizes achieving higher performance over real-time inference, unlike fields such as NLP where online responsiveness is critical. Furthermore, we emphasize that Spotscape without the prototypical contrastive loss (PCL) is computationally faster and outperforms most baselines. However, it shows some performance degradation, as reported in Figure 7. This makes it a viable alternative when faster inference is required, albeit with a trade-off in performance.

---

> ### Author Response · Authors · 2024-11-23
> **(4/N) Author Rebuttal**
>
> > The author mentions multiple times the statement that "In biological systems, cells that are spatially close often exhibit strong feature similarity, regardless of their functional characteristics." Is there any literature to justify this? From my knowledge, this statement is misleading. The similarity only appears at the spot level, while at the cell level, for example, the Xenium data, the cell types locally close to each other are actually quite heterogeneous. Tumor micro-environment is one of the few exceptions where the local cells tend to share similar characteristics. I hope the authors are rigid when making statements about biology.
>
> Thank you for your insightful observation, and I appreciate your domain knowledge in this area.
>
> I acknowledge that my original statement may have overstated the circumstances. At the cellular level, features are indeed heterogeneous enough to reflect distinct functional characteristics, as you noted.
> However, due to factors such as cellular interactions and the continuous nature of biological systems, gene expression values tend to vary smoothly along spatial coordinates. This results in spatially proximate spots often exhibiting relatively higher feature similarity compared to more distant spots.
>
> To address your valid concern, I have revised the statement to ensure clarity and biological accuracy: **"Biological systems exhibit a continuous nature, where gene expression values vary smoothly along spatial coordinates. This continuity leads to feature similarities between neighboring spots, influenced both by their spatial proximity and functional characteristics."**
>
> Thus, this statement still highlights the necessity of incorporating global context to effectively capture functional characteristics.
>
> Additionally, I also changed the same statement in lines 70-74 as well.
>
>
> > Meanwhile, statements are causing confusion like this "However, these approaches fall short in obtaining qualified spot representations, particularly for those located around the boundary of cell type clusters, as they heavily emphasize spatially local spots that have minimal feature differences from an anchor node." It mentions "cell type clusters" and "local spots" in the same time, while spots typically contain heterogeneous cell types, and they correspond to spatial domain instead of cell type clusters.
>
> I apologize for the confusion caused by the imprecise use of terminology. The statement you mentioned did not clearly distinguish between "cells" and "spots." I have revised the text, replacing "cell type clusters" with "spatial domains" to better reflect the intended meaning.

---

> ### Author Response · Authors · 2024-11-23
> **(5/N) Author Rebuttal**
>
> > "Prototypical Contrastive Learning" aims to enhance clustering and rare cell-type classification (I'm not sure how this benefits spot-level tasks). The idea is essentially similar to many previously established prototypical losses [1,2,etc]. However, I did not see any references in the paper. Should authors cite those methods or justify why they need to propose a different method?
>
> Thank you for pointing out this oversight regarding citations.
>
> You are correct that the Prototypical Contrastive Learning (PCL) used in our method aligns with the original PCL formulation. We sincerely apologize for omitting these critical references [1,2,3] in the initial submission. We have now included the relevant citations in the revised manuscript in line 249.
>
> [1] Prototypical Contrastive Learning of Unsupervised Representations, ICLR 2021
> [2] Population-level integration of single-cell datasets enables multi-scale analysis across samples, Nature Methods
> [3] Deep single-cell RNA-seq data clustering with graph prototypical contrastive learning, Bioinformatics.

---

> ### Author Response · Authors · 2024-11-23
> **(N/N) Author Rebuttal**
>
> > The authors proudly state that "instead of relying on any predictor or stop gradient techniques, Spotscape simplifies the training procedure by employing a reconstruction loss". To me, it is not clear how the reconstruction loss provides a better solution than the other options. It seems like the authors just blindly chose a shortcut for the implementation.
>
> Thank you for pointing out the need for clarification regarding our choice of reconstruction loss in Spotscape.
>
> Siamese networks typically require mechanisms to avoid degenerate solutions. While approaches like stop-gradient techniques used in BYOL and BGRL are effective, we opted for a reconstruction module instead.
>
> The reason is that, in this domain, the reconstructed output itself can serve valuable purposes by imputing and denoising raw transcriptomics value. To illustrate this, we evaluated the imputation performance in Figure 28 of Appendix I and demonstrated that Spotscape achieves superior results in imputing missing values.
>
> Furthermore, this denoised output can be used for marker gene identification. In Figure 29, we demonstrate that known marker genes in brain cortex layers, which are not highly expressed in the raw data, become distinguishable in the reconstructed output. This highlights the potential of the reconstruction component to enable additional tasks.

---

> ### Author Response · Authors · 2024-11-30
> **Gentle Reminder**
>
> **Dear Reviewer K25U,**
>
> Thank you for your insightful review and for dedicating your time to provide valuable feedback. Based on your comments, we have clarified our statements. Furthermore, we have conducted additional experiments.
>
>
> * Additional Experiments:
> 	* Comprehensive hyperparameter selection.
> 	* Hyperparameter sensitivity analysis.
> 	* Time complexity analysis.
> * Clarifications and Updates:
> 	* Clarification of our framework with detailed pseudo-code
> 	* Citations of PCL
> 	* Corrected terminology, e.g., "cell type" revised to "spatial domain"
>
> We would greatly appreciate it if you could share any further questions or comments. We look forward to your reevaluation of our work.
>
> Best regards,
> The Authors

---

> ### Author Response · Authors · 2024-12-03
> **Final Reminder Regarding Rebuttal Discussion**
>
> I sincerely apologize for the repetitive reminder. I would like to kindly follow up on my previous responses to your comments. As the rebuttal phase concludes in a few hours, I want to ensure that I have thoroughly addressed all your concerns.
>
> If there are any additional points requiring clarification, I would be happy to address them before the discussion period ends.
>
> Thank you for your time and consideration.

---

### Official Review · Reviewer_8AMT · 2024-11-07

**Soundness:** 3
**Presentation:** 4
**Contribution:** 2
**Rating:** 6
**Confidence:** 5

**Summary:**

This paper introduces a framework called Spotscape, designed to improve spatial transcriptomics data representation for enhanced analysis of spatial domain identification, and multi-slice integration and alignment. Spotscape utilizes a graph autoencoder and incorporates various loss terms to optimize the learning of spot representations, thereby improving clustering and alignment. Applied to five datasets, the framework demonstrates its effectiveness in predicting spatial domains of brain layers and mouse embryos, aligning temporal spatial transcriptomics data, and integrating spatial data across different resolutions.

**Strengths:**

1. Current whole-transcriptomic spatial sequencing platforms, such as 10x Visium, face challenges with low spot resolution, where each spot may capture transcripts from tens of cells. This limits the accuracy of clustering spatial domains, like brain layers. Spotscape demonstrated notable improvements in domain prediction accuracy in the human dorsolateral prefrontal cortex and mouse embryo data.
2. Batch effects are a common challenge in integrating and aligning multi-slice spatial data, particularly for temporal data analysis. Spotscape showed superior performance in reducing batch effects across most datasets compared to other methods.
3. Ablation studies highlighted the importance of incorporating various loss terms in the proposed graph auto-encoder model, demonstrating their critical role in enhancing data representation quality.

**Weaknesses:**

1. While Spotscape demonstrated significant improvements in spatial domain identification across two Visium datasets(DLPFC and MTG), its performance on the Stereo-seq platform for mouse embryo profiling showed only modest gains compared to existing algorithms, such as domain detection with STAGATE (Fig. 3B) and batch correction with STAligner (Table 4). This difference may stem from the higher resolution of Stereo-seq (0.2 µm per spot) compared to Visium’s lower resolution (55 µm per spot). The higher resolution likely enhances cell type and spatial domain identification, potentially diminishing Spotscape's advantage. With rapid advancements in spatial transcriptomics profiling—improving resolution, gene coverage, and sensitivity—Spotscape must demonstrate strength across both low-resolution platforms like Visium and high-resolution platforms like Stereo-seq, seqFISH+, Pixel-seq, etc., to contribute significantly to spatial data analysis.
2. While the inclusion of prototypical contrastive learning loss in Spotscape may improve spatial domain prediction, it could potentially impact other downstream tasks, such as trajectory prediction across cell types or states.
3. Several parameters, such as learning rate, number of clusters (K), and lambda values for each loss term, appear hardcoded in the provided code. It is unclear how to optimize these parameters for new datasets and whether a comprehensive search for optimal values is required. Additionally, it raises questions about the time needed for this process, especially for large datasets containing 40–100 spatial transcriptomics samples.
4. It is worth noting that single-cell embedding models like scGPT and scFoundation, which are trained on millions of cells, have demonstrated superior performance in cell type clustering and data integration. A comparison between Spotscape and these pre-trained deep learning models on spatial transcriptomics data would be valuable to determine if Spotscape can outperform these models on the proposed tasks.

**Questions:**

In addition to the main weaknesses, there are a few minor points of clarification:
1. In lines 40–42, the manuscript states, "However, they fall short in predicting accurate clustering results due to the inherent noise in SRT data." Clustering does not inherently have a predefined "accuracy." It would be clearer to specify accuracy in terms of cell type or brain layer annotation.
2. In lines 48–49, the manuscript mentions that GNN methods have limitations, especially for spots around cell type cluster boundaries, and subsequently states that the STAGATE algorithm was developed to address this issue. However, STAGATE primarily addresses boundaries of brain layers, not cell types. Assigning cell types to Visium data is challenging, as each spot may contain multiple cells. Researchers often prefer cell type deconvolution or refer to clusters as "cell type-enriched" rather than assigning a single cell type to each spatial transcriptomics cluster for low resolution data.
3. In lines 71–73, the manuscript states, "where cells that are spatially close often exhibit strong feature similarity even when they serve different functions." To support this general statement, it would be helpful to reference a review paper rather than a figure to clarify whether this phenomenon is specific to the dataset used or is observed across multiple datasets. This may not universally apply to all datasets, as spatial transcriptomics profiling is widely used to detect cell type and tissue structure differences in various disease contexts.

---

> ### Author Response · Authors · 2024-11-23
> **(1/N) Author Rebuttal**
>
> Thank you so much for your thorough and insightful feedback!
>
> > While Spotscape demonstrated significant improvements in spatial domain identification across two Visium datasets(DLPFC and MTG), its performance on the Stereo-seq platform for mouse embryo profiling showed only modest gains compared to existing algorithms, such as domain detection with STAGATE (Fig. 3B) and batch correction with STAligner (Table 4). This difference may stem from the higher resolution of Stereo-seq (0.2 µm per spot) compared to Visium’s lower resolution (55 µm per spot). The higher resolution likely enhances cell type and spatial domain identification, potentially diminishing Spotscape's advantage. With rapid advancements in spatial transcriptomics profiling—improving resolution, gene coverage, and sensitivity—Spotscape must demonstrate strength across both low-resolution platforms like Visium and high-resolution platforms like Stereo-seq, seqFISH+, Pixel-seq, etc., to contribute significantly to spatial data analysis.
>
> Thank you for your valuable feedback and for pointing out our experiment's limitations. In our previous experiments, the results in Table 4 pertain to MTG data obtained from the Visium platform. The clustering results from high-resolution sequencing (subcellular level) are only presented in Table 2 (now updated to Table 3 in the revised version) and Figure 3. Although our proposed model shows the best performance on this data, we acknowledge your concern that Spotscape provides only modest improvements compared to the baseline method when applied to the higher-resolution Stereo-seq platform.
>
> To address this concern and strengthen our paper, we have conducted additional experiments using non-small cell lung cancer (NSCLC) data obtained from CosMX sequencing, which offers one of the highest subcellular resolutions. Our results demonstrate that Spotscape outperforms other baseline methods and shows significant improvements, with the exception of SpaceFlow. We believe this additional experiment highlights the strengths of Spotscape across both low- and high-resolution platforms, providing a more comprehensive evaluation of its capabilities in spatial transcriptomics analysis.

---

> ### Author Response · Authors · 2024-11-23
> **(2/N) Author Rebuttal**
>
> > While the inclusion of prototypical contrastive learning loss in Spotscape may improve spatial domain prediction, it could potentially impact other downstream tasks, such as trajectory prediction across cell types or states.
>
> Thank you for pointing out the potential concern regarding the impact of prototypical contrastive learning loss on trajectory inference tasks. We acknowledge that prototypical loss could reduce the continuity of the latent space, which might negatively affect trajectory prediction across cell types or states. However, in our model optimization process, the prototypical loss is weighted relatively low (i.e., $\lambda_{PCL}$ is searched in {0.0005, 0.001, 0.005, 0.01}) compared to other loss terms, thereby minimizing its influence on the overall latent space structure.
>
> To empirically address this concern, we performed trajectory inference tasks, as detailed in Appendix H. In the experiments shown in Figure 26, we quantitatively assess trajectory inference performance by computing the correlation between the pseudo-Spatiotemporal Map (pSM) values and gold standard spatial regions. Our results demonstrate that Spotscape achieves effective trajectory inference, indicating that the inclusion of prototypical loss does not adversely impact this task. However, as anticipated, we observed that setting $\lambda_{PCL}$ to a higher value does lead to a decline in performance on trajectory inference.
>
> We have also visualized the trajectory inference results and pSM maps in Figure 27 for better clarity.

---

> ### Author Response · Authors · 2024-11-23
> **(3/N) Author Rebuttal**
>
> > Several parameters, such as learning rate, number of clusters (K), and lambda values for each loss term, appear hardcoded in the provided code. It is unclear how to optimize these parameters for new datasets and whether a comprehensive search for optimal values is required.
>
> We apologize for not clarifying the hyper-parameter selection process, thereby inducing confusion about our reported experiment results.
>
> We acknowledge that some studies in this domain report the performance of baseline methods using their default hyperparameters, reflecting the unsupervised nature of tasks such as clustering or alignment.
> However, the datasets used in different studies often vary, and even when the same dataset is used, preprocessing steps can differ, potentially leading to underestimation of baseline performance when default parameters are applied.
> To ensure a fair comparison, we performed a comprehensive hyperparameter search for both our model and the baseline models, optimizing for test performance.
> This approach allows us to provide a more accurate and fair evaluation of all methods.
> Additionally, we search all balance parameters and learning rates for each baseline method, including any additional parameters required for certain baselines. A more detailed description of the search space and finally selected parameters are provided in Appendix Section E.1.
>
> Furthermore, to address potential concerns that our model's performance depends on hyperparameters tuned using test performance, we conducted an extensive sensitivity analysis and also suggested a reasonable unsupervised hyperparameter selection strategy.
> From the sensitivity analysis in Appendix Section D, we observed that, aside from the learning rate, all hyperparameters of Spotscape are robust in reasonable space, indicating that their selection has minimal impact on performance.
> Building on this insight, in Appendix Section E.2, we fixed all hyperparameters except the learning rate and selected its optimal learning rate from the set {0.001, 0.0005, 0.0001, 0.00005, 0.0000} based on the silhouette score, which can be obtained unsupervised manner.
> As a result, this strategy requires a total of $5 \times$ the running time. For example, while a single DLPFC sample takes 64.4 seconds to run Spotscape, it requires 322 seconds to find the optimal learning rate.
> The results of this search are reported in Table 11, with the corresponding silhouette scores for the DLPFC data shown in Figure 21.
> Furthermore, in Figure 22, we compared the performance achieved using these unsupervised-selected hyperparameters with that obtained from supervised NMI-based selection and found that both approaches yield similar results.
> This demonstrates the effectiveness of our search strategy and confirms the robustness of hyperparameter sensitivity.
>
> Additionally, we conducted a sensitivity analysis regarding the number of clusters during K-means clustering, as shown in Figure 20. In this analysis, we set the number of clusters K to the ground truth number of spatial domains and performed clustering with values of K−2, K−1, K, K+1, and K+2. Our results show that the performance does not degrade significantly when the number of clusters is slightly incorrect.
>
> > Additionally, it raises questions about the time needed for this process, especially for large datasets containing 40–100 spatial transcriptomics samples.
>
> To address this concern, we synthesized datasets containing 40, 60, 80, and 100 spatial transcriptomics samples by replicating the DLPFC dataset to measure running time. The results are summarized in the table below:
>
> |  Running time (sec) | **40**   | **60**   | **80**   | **100**  |
> |---------------------|----------|----------|----------|----------|
> | Spotscape           | 18,985.50 | 31,667.34 | 43,357.98 | 52,075.94 |
> | Spotscape w/o $L_{PCL}$          | 16,326.55 | 26,905.04 | 34,291.23 | 41,880.16 |
>
> From these results, we observed that batch training enables handling large-scale datasets, completing the process within reasonable time frames. However, we acknowledge that GNN-based models, including Spotscape and existing methods, are less suitable for fast inference on datasets of this magnitude. Addressing this limitation could be a promising direction for future work.

---

> ### Author Response · Authors · 2024-11-23
> **(4/N) Author Rebuttal**
>
> > It is worth noting that single-cell embedding models like scGPT and scFoundation, which are trained on millions of cells, have demonstrated superior performance in cell type clustering and data integration. A comparison between Spotscape and these pre-trained deep learning models on spatial transcriptomics data would be valuable to determine if Spotscape can outperform these models on the proposed tasks.
>
> Thank you for your suggestion to compare Spotscape with pre-trained models such as scGPT and scFoundation, which have demonstrated strong performance in cell type clustering and data integration. To address this, we conducted experiments with scGPT, a representative model, using the DLPFC slice 151673.
>
> |           |  Silhouette |     ARI     |     NMI     |
> |-----------|:-----------:|:-----------:|:-----------:|
> |    SEDR   | 0.24 (0.03) | 0.36 (0.08) | 0.49 (0.08) |
> |  STAGATE  | 0.18 (0.02) | 0.37 (0.04) | 0.55 (0.03) |
> |   SpaCAE  | 0.35 (0.05) | 0.21 (0.01) | 0.37 (0.01) |
> | SpaceFlow | 0.43 (0.01) | 0.42 (0.06) | 0.57 (0.05) |
> |  GraphST  | 0.29 (0.01) | 0.20 (0.02) | 0.34 (0.03) |
> |   scGPT   | 0.03 (0.00) | 0.19 (0.00) | 0.31 (0.00) |
> | Spotscape | **0.46 (0.02)** | **0.47 (0.03)** | **0.62 (0.02)** |
>
> The results from scGPT showed significantly lower performance compared to Spotscape and other baselines. We believe this is primarily due to scGPT's inability to incorporate spatial information, which is crucial for tasks in spatial transcriptomics. This highlights the importance of spatial context in enhancing model performance in this domain.

---

> ### Author Response · Authors · 2024-11-23
> **(N/N) Author Rebuttal**
>
> > In lines 40–42, the manuscript states, "However, they fall short in predicting accurate clustering results due to the inherent noise in SRT data." Clustering does not inherently have a predefined "accuracy." It would be clearer to specify accuracy in terms of cell type or brain layer annotation.
>
> Thank you for pointing out the use of an inappropriate term. we have replaced 'clustering' with 'spatial domain' to clarify the statement.
>
> > In lines 48–49, the manuscript mentions that GNN methods have limitations, especially for spots around cell type cluster boundaries, and subsequently states that the STAGATE algorithm was developed to address this issue. However, STAGATE primarily addresses boundaries of brain layers, not cell types. Assigning cell types to Visium data is challenging, as each spot may contain multiple cells. Researchers often prefer cell type deconvolution or refer to clusters as "cell type-enriched" rather than assigning a single cell type to each spatial transcriptomics cluster for low resolution data.
>
> Thank you for pointing out the inconsistency. To clarify, I have changed 'the boundary of cell types' to 'the boundary of different spatial domains,' as STAGATE primarily addresses the boundaries of spatial domains (such as brain layers), rather than cell types.
>
> > In lines 71–73, the manuscript states, "where cells that are spatially close often exhibit strong feature similarity even when they serve different functions." To support this general statement, it would be helpful to reference a review paper rather than a figure to clarify whether this phenomenon is specific to the dataset used or is observed across multiple datasets. This may not universally apply to all datasets, as spatial transcriptomics profiling is widely used to detect cell type and tissue structure differences in various disease contexts.
>
> Thank you for pointing out the need for clarification. We acknowledge that our previous statement, **“cells that are spatially close often exhibit strong feature similarity even when they serve different functions,”** was overstated, as not all spatial transcriptomics datasets show such high feature similarity across different biological contexts.
>
> To address this, we revised the statement as follows:
> **“We argue that learning attention weights in SRT data is particularly challenging due to the continuous nature of biological systems, where gene expression values tend to vary smoothly along spatial coordinates [1,2,3,4]. This inherent continuity can, in some cases, complicate the distinction between different spatial domains.”**
>
> This revised statement avoids making a universal claim. Instead, it highlights that smooth variation in gene expression across spatial domains, supported by existing references, can pose challenges in certain datasets without overgeneralizing.
>
> [1] Continuous variation within cell types of the nervous system, Trends in Neurosciences.
> [2] A repeated molecular architecture across thalamic pathways. Nature neuroscience.
> [3] Continuum of gene-expression profiles provides spatial division of labor within a differentiated cell type. Cell systems.
> [4] Single-cell co-expression analysis reveals that transcriptional modules are shared across cell types in the brain. Cell systems.

---

> ### Author Response · Authors · 2024-11-30
> **Gentle Reminder**
>
> **Dear Reviewer 8AMT,**
>
> Thank you once again for your thoughtful and detailed feedback on our manuscript. As per your suggestions, we have added clarifications to our statements and conducted additional experiments.
>
> * Additional Experiments:
>     * Performance evaluation on high-resolution platforms (CosMX sequencing and Stereo-seq).
>     * Analysis of PCL’s impact on trajectory prediction.
>     * Comprehensive hyperparameter selection and sensitivity analysis.
>     * Running time evaluation for large-scale datasets (40–100 samples).
>         Comparison with foundation models (scGPT).
> * Clarifications and Updates:
>     * Refined statements on spatial domains, clustering, and feature similarities for clarity and precision.
>     * Corrected terminology, e.g., "cell type" revised to "spatial domain"
>
> We kindly remind you to review the updated manuscript and share any additional questions or concerns. We sincerely appreciate your insights and look forward to your reevaluation.
>
> Best regards,
> The Authors

---

> ### Author Response · Authors · 2024-12-03
> **Final Reminder Regarding Rebuttal Discussion**
>
> I sincerely apologize for the repetitive reminder. I would like to kindly follow up on my previous response to your comments. As the rebuttal phase concludes in a few hours, I want to ensure that I have thoroughly addressed all your concerns.
>
> If there are any additional points requiring clarification, I would be happy to address them before the discussion period ends.
>
> Thank you for your time and consideration.

---

### Meta-Review · Area_Chair_KgzB · 2024-12-19

**Metareview:**

This submission focuses on analyzing Spatially Resolved Transcriptomics (SRT) data and introduces Spotscape for enhanced representation learning. Spotscape utilizes a graph auto-encoder and incorporates various loss terms to optimize the learning of spot representations. The authors reported empirical results on multiple datasets to validate its effectiveness in improving clustering and alignment across different resolutions.

Most of the reviewers agreed that the demonstrated empirical performance by Spotscape has shown to improve SRT data analysis. Remaining concerns include the incremental methodological contributions, computational complexity and scalability, generalizability to other SRT platforms, the effectiveness of trajectory or perturbation prediction downstream tasks besides the reported results dependent mostly on representation learning from training data.

Given these remaining concerns, this submission with its current content is more suitable for other high impact avenues where the readership is more inclined towards bioinformatics data analysis.

The authors may need to provide more comprehensive experiments considering addressing the remaining concerns to further improve the quality of the paper. For example, the results with CosMX sequencing during the rebuttal on other platforms, should be included and expanded to better understand the limitations of the proposed method. More comprehensive results using other datasets on prediction tasks can also further strengthen the presented work.

**Additional Comments On Reviewer Discussion:**

During the rebuttal, the authors have provided clarifications and additional results to address the reviewers' concerns on sensitivity analysis and generalizability to other platforms and downstream tasks. Some borderline reviewers have remaining concerns on incremental methodological contributions, computational complexity and scalability, and generalizability.

---

### Decision · Program_Chairs · 2025-01-22

Reject